# Smoothed Differentiation Efficiently Mitigates Shattered Gradients in Explanations

**Adrian Hill**[1 2]    **Neal McKee**[3 4]    **Johannes Maeß**[1 2]
**Stefan Blücher**[1 2]    **Klaus-Robert Müller**[1 2 5 6]

[1]BIFOLD – Berlin Institute for the Foundations of Learning and Data, Berlin, Germany
[2]Machine Learning Group, TU Berlin, Berlin, Germany
[3]Bernstein Center for Computational Neuroscience, Berlin, Germany
[4]TU Berlin, Berlin, Germany
[5]Department of Artificial Intelligence, Korea University, Seoul, Korea
[6]Max Planck Institut für Informatik, Saarbrücken, Germany

hill@tu-berlin.de   neal@bccn-berlin.de   maess@tu-berlin.de
bluecher@tu-berlin.de   klaus-robert.mueller@tu-berlin.de

## Abstract

Explaining complex machine learning models is a fundamental challenge when developing safe and trustworthy deep learning applications. To date, a broad selection of explainable AI (XAI) algorithms exist. One popular choice is SmoothGrad, which has been conceived to alleviate the well-known shattered gradient problem by smoothing gradients through convolution. SmoothGrad proposes to solve this high-dimensional convolution integral by sampling – typically approximating the convolution with limited precision. Higher numbers of samples would amount to higher precision in approximating the convolution but also to higher computing demand, therefore in practice only few samples are used in SmoothGrad. In this work we propose a well founded novel method *SmoothDiff* to resolve this tradeoff yielding a *speedup of over two orders of magnitude*. Specifically, *SmoothDiff* leverages automatic differentiation to decompose the expected values of Jacobians across a network architecture, directly targeting only the non-linearities responsible for shattered gradients and making it easy to implement. We demonstrate SmoothDiff's excellent speed and performance in a number of experiments and benchmarks. Thus, SmoothDiff greatly enhances the usability (quality and speed) of SmoothGrad – a popular workhorse of XAI.

## 1   Introduction

XAI has in the past years become a popular resource to further a user's understanding of various machine learning models such as neural networks, kernel methods, LSTMs and transformers, see e.g. [1]. Among others, XAI allows to debug models, scrutinize data sets and find Clever-Hans effects in supervised and unsupervised models [2, 3]. Post-hoc explanation methods, see [1], can e.g. be based on gradients [4, 5, 6], first- and higher-order Taylor expansions around meaningful root points [7, 8, 9, 10], and Shapley values [11, 12].

In this work we will place our focus on the inner workings of a classifier by inspecting the gradient of its prediction with respect to its input features. Usually, the focus is on the gradient $\nabla f_c(\boldsymbol{x})$ of the maximally activated output logit, however any other logit can be inspected as well, resulting in an explanation for that respective class (see Appendix A for notation). We limit ourselves to the fixed-model regime, in which the model cannot be retrained for the purpose of interpretability.

39th Conference on Neural Information Processing Systems (NeurIPS 2025).

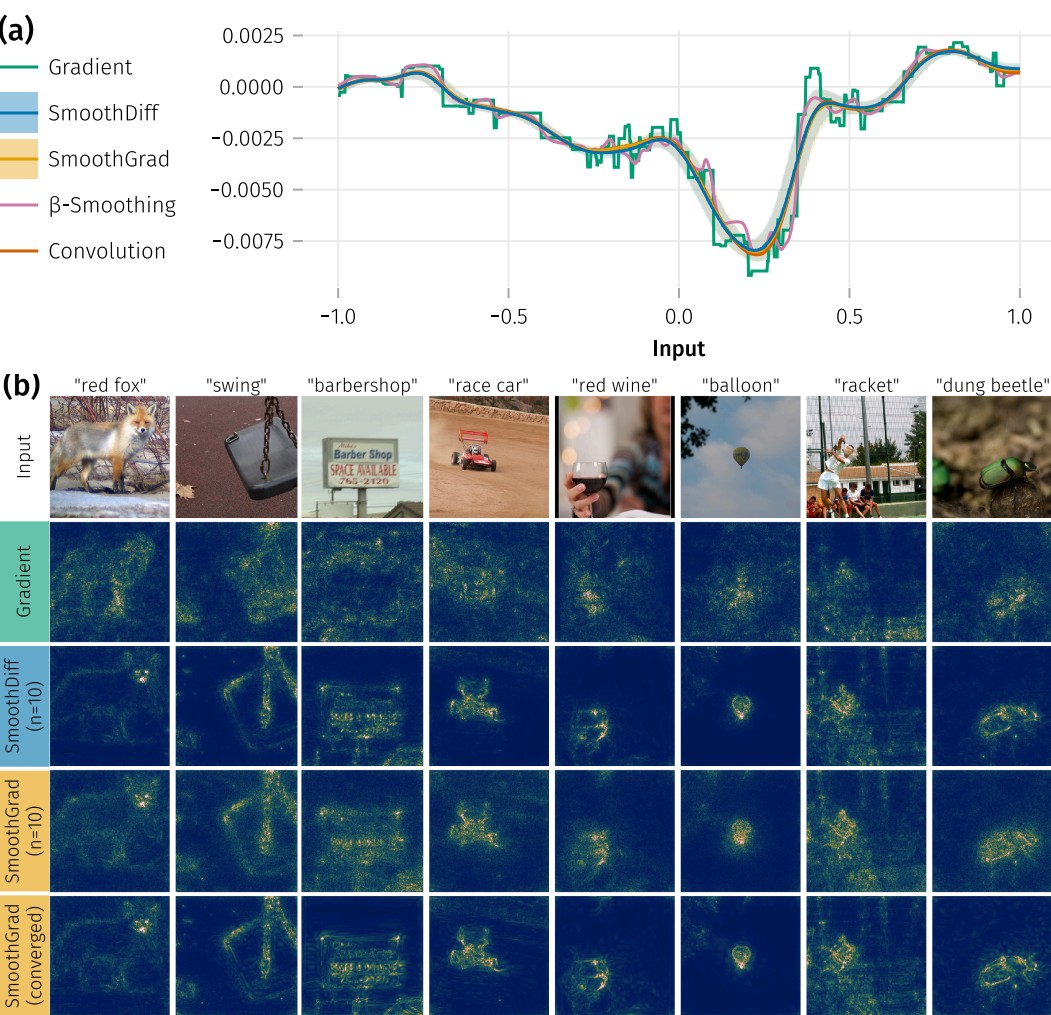

Figure 1: **(a)** Visualization of the shattered gradient problem on a 1D deep ReLU network. Gaussian convolution removes high-frequency components of the shattered gradient and is well approximated by both SmoothDiff and SmoothGrad. **(b)** On high-dimensional ImageNet models, SmoothGrad suffers from low sample efficiency, whereas at just 10 samples, SmoothDiff's level of noise reduction matches a fully converged SmoothGrad. See Appendix B for implementation details.

## 1.1 Preliminaries

**Shattered gradient problem**. With growing depth of neural networks, the vector field of gradients over the input space $\mathcal{X}$ gets increasingly noisy, as shown in [13]. For small changes in inputs $x \in \mathcal{X}$, resulting gradient vectors $\nabla f_c(x)$ vary strongly, resembling white noise. This phenomenon has been termed the *shattered gradient problem* (SGP), namely the SGP renders them uninformative due to a bad signal to noise ratio, as shown in Figure 1: For a 1-dimensional toy example we observe that the SGP results in large step-wise fluctuations of the gradient; also for gradient explanations in vision models (here VGG-19 [14]), the SGP leads to highly noisy heatmaps.

**Gaussian convolutions**. Since the SGP closely resembles white noise in deep networks, it motivates the use of a Gaussian filter in the input domain to suppress high-frequency signals: if images are perceptually robust to small amounts of noise, explanations should also be stable with respect to minor input variations. This implies that explanations should not vary rapidly and have little high-frequency content in the input domain. Mathematically, the desired Gaussian filter operation corresponds to a convolution with a Gaussian kernel $g$ over the input domain $\mathcal{X}$. This in turn is equal to the expected value

$$(\nabla f_c * g)(\boldsymbol{x}) = \int_{\mathcal{X}} \nabla f_c(\hat{\boldsymbol{x}}) g(\boldsymbol{x} - \hat{\boldsymbol{x}}) d\hat{\boldsymbol{x}} = \mathbb{E}_{\hat{\boldsymbol{x}} \sim \mathcal{N}_g}[\nabla f_c(\hat{\boldsymbol{x}})] \,, \tag{1}$$

where $\mathcal{N}_g$ is the Normal distribution analogous to the kernel $g$ in terms of covariances [15]. The variance of this distribution represents a trade-off between the strength of the smoothing and using information from regions of the input space that are no longer relevant to the actual input. It should be chosen large enough to aggregate gradient information over similar inputs, but small enough to not change the classification or subjective qualitative nature of the input to humans.

**SmoothGrad**. While the convolution in (1) is generally intractable or at least unfeasible, it can be approximated via sampling [6]. Using $n$ samples $\hat{\boldsymbol{x}}_i$ drawn from $\mathcal{N}_g$, we obtain the definition of SmoothGrad (SG) as

$$\mathrm{SG}_n(\boldsymbol{x}) = \frac{1}{n} \sum_{i=1}^{n} \nabla f_c(\hat{\boldsymbol{x}}_i) \,. \tag{2}$$

Since this corresponds to an averaging of input gradients over noisy inputs, the computational cost of SG grows linearly in the amount of samples, requiring $n$ forward and $n$ backward passes through the model, with a suggested sample size of $n = 50$ [6]. While SG converges to the convolution (1) in the infinite sample limit, it is well-known that computationally tractable amounts of samples may not be sufficient in practice, say, for ImageNet samples, as is visually apparent in the heatmaps of Figure 1 (b) and demonstrated in Section 3.1. Due to this, SmoothGrad is commonly seen as a *perturbation-based* method instead of one based on Gaussian convolution.

**Gradient explanations rewritten as factorized Jacobians**. For any function $f : \mathbb{R}^n \to \mathbb{R}^m$, the gradient of the $c$-th output corresponds to the $c$-th row of the Jacobian $\boldsymbol{J}_f(\boldsymbol{x})$. For a deep neural network $f = f^N \circ f^{N-1} \circ ... \circ f^1$ composed of $N$ functions $f^n$, this gradient can therefore be derived via the chain rule as

$$\begin{aligned}
\nabla f_c(\boldsymbol{x})^T = \left[\boldsymbol{J}_f(\boldsymbol{x})\right]_{c,:} &= \boldsymbol{e}_c^T \cdot \boldsymbol{J}_f(\boldsymbol{x}) \\
&= \boldsymbol{e}_c^T \cdot \boldsymbol{J}_{f^N}(\boldsymbol{a}^N) \cdot \boldsymbol{J}_{f^{N-1}}(\boldsymbol{a}^{N-1}) \cdot ... \cdot \boldsymbol{J}_{f^1}(\boldsymbol{x}) \,,
\end{aligned} \tag{3}$$

where we denote the input activation of the $n$-th function as $\boldsymbol{a}^n$, such that the input $\boldsymbol{x} = \boldsymbol{a}^1$ and $\boldsymbol{a}^{n+1} = f^n(\boldsymbol{a}^n)$. We would like to stress that such gradients can efficiently be computed using reverse mode automatic differentiation (AD), sequentially accumulating the *vector-Jacobian products* (VJPs) on the right-hand side of (3) from the left to the right [16, 17, 18]. Since this computation requires knowledge of all activations $\boldsymbol{a}^N$ to $\boldsymbol{a}^1$, a forward pass through the model must first be computed, saving all activations in memory for the subsequent backward pass. By writing the forward pass in AD frameworks such as PyTorch [19] and JAX [20], gradients explanations (3) can be computed up to numerical accuracy without requiring any additional code.

## 1.2 Contributions

In this work, we provide the following key contributions:

1. We motivate Gaussian convolutions in embedding space as a solution to the shattered gradient problem [13], emphasize that SmoothGrad numerically approximates said convolution in the infinite sample limit, and demonstrate that commonly used SmoothGrad sample sizes are typically insufficient.
2. We introduce SmoothDiff based on AD, which is both faster and more sample efficient than SmoothGrad in approximating Gaussian convolutions. We provide experimental benchmarks and clear guidelines on how to implement SmoothDiff for arbitrary differentiable model architectures.
3. We demonstrate that SmoothDiff directly and efficiently addresses the shattered gradient problem.

## 1.3 Related work

*NoiseGrad* is a variant of SG that computes an average of gradient explanations over perturbed model weights instead of perturbed inputs [21]. It is therefore not connected to input space convolutions.

Perturbing both inputs and model weights results the *FusionGrad* method. In [22], a low-pass filter is applied to gradient explanations in image space. Since the smoothing is not applied in input space, this approach does not directly address the SGP and runs into the risk of removing meaningful high-frequency spatial information from heatmaps. In [23], the high curvature of neural network decision boundaries and how it can be exploited to manipulate explanations is explored. The proposed $\beta$-*Smoothing* method smoothes the decision boundary by replacing ReLUs with Softplus activations on the backward pass. As we show in Appendix G, this is a good approximation of a Gaussian convolution in the case of a single-layer neural network. However, as shown in Figure 1(a), the approximation error grows with network depth, as after the first layer, subsequent activations may not be Gaussian anymore.

## 2 SmoothDiff

We now propose SmoothDiff, a computationally efficient approximation of the Gaussian convolution. Inserting the factorized neural network gradient (3) into the convolution (1), we obtain

$$(\nabla f_c * g)(\boldsymbol{x}) = \boldsymbol{e}_c^T \cdot \mathbb{E}_{\hat{\boldsymbol{x}} \sim \mathcal{N}_g} \left[ \boldsymbol{J}_{f^N}(\hat{\boldsymbol{a}}^N) \cdot \boldsymbol{J}_{f^{N-1}}(\hat{\boldsymbol{a}}^{N-1}) \cdot ... \cdot \boldsymbol{J}_{f^1}(\hat{\boldsymbol{x}}) \right] , \tag{4}$$

in which all intermediate activations $\hat{\boldsymbol{a}}^i$ deterministically depend on the random variable $\hat{\boldsymbol{x}}$. By neglecting cross-covariances between Jacobians, this expected value of a product of Jacobians can be approximated as a product of expected values of Jacobians, resulting in the definition of SmoothDiff:

$$\text{SD}(\boldsymbol{x}) = \boldsymbol{e}_c^T \cdot \underbrace{\mathbb{E}_{\hat{\boldsymbol{a}}^N} \left[ \boldsymbol{J}_{f^N}(\hat{\boldsymbol{a}}^N) \right]}_{=:\widehat{\boldsymbol{J}_{f^N}}} \cdot ... \cdot \underbrace{\mathbb{E}_{\hat{\boldsymbol{x}}} \left[ \boldsymbol{J}_{f^1}(\hat{\boldsymbol{x}}) \right]}_{=:\widehat{\boldsymbol{J}_{f^1}}} . \tag{5}$$

Note that this definition is based on the gradient decomposition (3), however, here we propose to *replace* Jacobians $\boldsymbol{J}_{f^i}$ by the expected values of Jacobians $\widehat{\boldsymbol{J}_{f^i}}$.

### 2.1 Jacobian factorization

The chain rule can be used to factorize neural network Jacobians to arbitrary levels of granularity: from layer-wise Jacobians down to derivatives of elementary functions like addition and multiplication. For SmoothDiff, this raises the question of optimal Jacobian factorization (5), as this choice affects which cross-covariance terms are neglected, trading an approximation bias for increased sample efficiency and computational performance. While in theory many options arise, in practice the choice of factorization for a given model architecture is typically straightforward.

We provide the following intuition: Since neural networks $f : \mathbb{R}^n \to \mathbb{R}^m$ are universal function approximators, their Jacobians–which correspond to linearizations of said universal function–can be arbitrary $m \times n$ matrices. Instead of estimating such "unconstrained" network Jacobians directly (2), sample efficiency can be improved by decomposing the network down to functions with constrained Jacobian structure. This includes:

1. **Linear and affine functions**. Since their Jacobians are constant, $\hat{\boldsymbol{J}} = \boldsymbol{J}$ is known.
2. **Functions with sparse Jacobians** (i.e. containing mostly zeros). Examples include vectorized scalar functions, such as element-wise activation functions, whose Jacobians are diagonal matrices: Since the $i$-th output only depends on the $i$-th input, $\hat{J}_{i,j} = J_{i,j} = \frac{\partial f_i}{\partial x_j} = 0$ for $i \neq j$.
3. **Functions with bounded partial derivatives** $\frac{\partial f_i}{\partial x_j}$. This property appears in common activation functions. For ReLU, softplus and $\tanh$, it holds that $0 \leq \frac{\partial f}{\partial x} \leq 1$.
4. **Functions with a closed-form analytical solution to the expected Jacobian $\hat{\boldsymbol{J}}$**. Point 1 is a special case of this.

Applied to common network architectures, these principles discourage us from naively applying the factorization on a layer-wise basis. Instead, element-wise activation functions should be separated from otherwise affine functions. The Jacobian of a dense layer $f(\boldsymbol{x}) = \sigma(g(\boldsymbol{x})) = \sigma(\boldsymbol{W}\boldsymbol{x} + \boldsymbol{b})$ should therefore be decomposed as

$$\boldsymbol{J}_f = \boldsymbol{J}_\sigma \cdot \boldsymbol{J}_g = \boldsymbol{J}_\sigma \cdot \boldsymbol{W} . \tag{6}$$

In other words: instead of estimating the dense matrix $\widehat{\boldsymbol{J}}_f$ containing $m \cdot n$ entries, where $n$ is the input and $m$ the output dimensionality, we should leverage our knowledge of the constant Jacobian $\widehat{\boldsymbol{J}}_g = \boldsymbol{J}_g = \boldsymbol{W}$ and only estimate the $m$ bounded entries in the sparse diagonal matrix $\widehat{\boldsymbol{J}}_\sigma$, resulting in a higher sample efficiency. We give an implementation example of this in the following.

## 2.2 Example implementation

SmoothDiff can be implemented as a *modified-backprop method*,[1] replacing the sequential computation of VJPs (3) by what we term vector-*expected-Jacobian* products (VEJPs) in (5), sequentially evaluating $\boldsymbol{v}^T \widehat{\boldsymbol{J}}_{f^n}$ on a single backward pass. For linear and affine functions with expected Jacobian $\widehat{\boldsymbol{J}} = \boldsymbol{J}$, it follows that $\boldsymbol{v}^T \widehat{\boldsymbol{J}} = \boldsymbol{v}^T \boldsymbol{J}$. Applying SmoothDiff to such functions requires no additional code, as common AD frameworks such as PyTorch and JAX already implement corresponding VJPs. SmoothDiff only requires implementing VEJPs for functions in the model which are non-linear with respect to their input, such that $\widehat{\boldsymbol{J}} \neq \boldsymbol{J}$. If the model contains functions for which $\widehat{\boldsymbol{J}}$ has no analytical solution, sampling estimates can be obtained by computing $n$ forward passes through the model. As a motivating example, we now discuss how to implement a VEJP for ReLUs $\sigma$. This should provide a blueprint on how to implement SmoothDiff on a variety of current and future model architectures.

As discussed in the previous section, the Jacobian $\boldsymbol{J}_\sigma$ of any element-wise activation function is a diagonal matrix. For ReLUs specifically, the diagonal entries correspond to

$$J_{\sigma_{i,i}}(\boldsymbol{a}) = \mathbb{1}_{x \geq 0}(a_i) := \begin{cases} 1 \text{ if } a_i \geq 0 \\ 0 \text{ else} \end{cases}, \tag{7}$$

where $\mathbb{1}_{x \geq 0}$ is the indicator function returning one for inputs greater or equal zero. From this, we can derive the expected Jacobian $\widehat{\boldsymbol{J}}_\sigma$ as

$$\widehat{\boldsymbol{J}}_{\sigma_{i,i}} = \underset{\hat{\boldsymbol{a}}}{\mathbb{E}} \Big[ J_{\sigma_{i,i}}(\hat{\boldsymbol{a}}) \Big] = \lim_{n \to \infty} \underbrace{\frac{1}{n} \sum_{j=1}^{n} \mathbb{1}_{x \geq 0}(a_i)}_{=:p_i}, \tag{8}$$

where the right-hand side corresponds to the expected ratio of positive activations. A sampling approximation of this term can be obtained by computing a finite amount of $n$ forward passes through the model with inputs $\hat{\boldsymbol{x}}$ drawn from $\mathcal{N}_g$, while keeping track of the count of non-zero activations $p_i$. For a given vector $\boldsymbol{v}$, this results in the VEJP

$$\Big[ \boldsymbol{v}^T \widehat{\boldsymbol{J}}_\sigma \Big]_i = v_i \widehat{\boldsymbol{J}}_{\sigma_{i,i}} = v_i \frac{p_i}{n}, \tag{9}$$

such that the SmoothDiff explanation (5) can be computed in a single matrix-free backward pass. An analogous approach can be taken to estimate $\widehat{\boldsymbol{J}}$ of max pooling layers, whose entries correspond to binary indicator functions of maximally activated inputs, requiring a sampling estimate of the ratio of activations with maximum value. Pseudocode implementation for both ReLUs and max pooling layers are given in Appendix I. While our exposition emphasized sequential models for the sake of simplicity, more complex, branching models such as ResNets [24] are automatically handled by AD systems. In fact, implementing the two aforementioned VEJPs is all it takes to run SmoothDiff on common vision models such as VGG and ResNets.

## 2.3 Fallback VEJP

SmoothDiff can be applied to arbitrary differentiable models. For any function (e.g. an arbitrary layer), a fallback VEJP can be implemented that samples $n$ forward passes, computes the full Jacobian for each sample, and approximates the expected value by keeping track of the running mean. The naive computation of a Jacobian requires either as many VJPs as the function has outputs (reverse mode) or as many *Jacobian-vector products* (JVPs) as it has inputs (forward mode). If the Jacobian exhibits sparsity, *automatic sparse differentiation* [17, 25] can drastically reduce the number of required VJPs/JVPs (see [26, 27] for applications in deep learning).

---

[1] The term *modified-backprop method* is a misnomer based on implementation details. Backpropagation is a special case of reverse mode AD, while Jacobian transformations (such as our expected Jacobians) can in many cases also be implemented for forward mode AD systems. Refer to [17] and [15] for an overview of AD techniques.

For functions with diagonal Jacobians (e.g. activation functions) this fallback VEJP is efficient, only requiring a single VJP per sample. For large and dense Jacobians however, it requires a lot of compute and memory, negating SmoothDiff's performance benefits over SG.

## 2.4 Computational performance and other properties

As we will demonstrate in Section 3.1, estimating factorized Jacobians according to Section 2.1 is significantly more sample efficient than estimating the Jacobian of the entire neural network. In practice, SmoothDiff therefore requires an order of magnitude fewer samples than SG to achieve a visually comparable (see Section 3.3 and Appendix K) amount of smoothing or noise reduction.

Additionally, for single class explanations of sample size $n$, SmoothDiff only requires $n$ forward and a single backward pass, whereas SG requires $n$ forward and $n$ backward passes. Denoting the unit time complexity of a single forward pass $f(\boldsymbol{x})$ by $\tau$, the corresponding backward pass has been shown to be in the same order of time complexity $O(\tau)$ [28], and we (conservatively) estimate their costs to be equal. For single-class explanations with $n$ samples, SmoothDiff yields a stark improvement requiring $n + 1$ (forward or backward) passes in comparison to SG's $2n$.

The largest performance gains occur when multiple explanations need to be computed for a given input, e.g. multi-class explanations and explanations with respect to concepts or filters that restrict the propagation through specific neurons. This is required for spectral analysis of explanations [29], concept relevance propagation [30], and higher-order explanation techniques [9, 10, 31]. So far, such methods have predominantly used the highly efficient LRP method [7, 8]. Assuming explanations of $k$ classes (or concepts/filters), LRP requires only a single forward pass and $k$ backward passes. In contrast, SG only estimates class-dependent smoothed gradients (2) and needs to compute $n$ forward and $n$ backward passes for each of the $k$ classes, resulting in a total of $2nk$ passes.
Since SmoothDiff estimates the full smoothed Jacobian (5) using matrix-free operators (VEJPs), it can be applied at a similar time complexity to LRP. Once VEJPs are estimated in $n$ forward passes, they don't need to be recomputed and only one backward pass per class (or concept/filter) is needed, resulting in a total of $n + k$ passes. Assuming an explanation of all $k = 1000$ ImageNet classes using $n = 50$ samples,[2] SmoothDiff yields a computational speedup of two orders of magnitude[3] over SG's approximation to Gaussian filtering and approaches three orders of magnitude including the increased sample efficiency demonstrated in Section 3.

In addition to convergence and performance improvements, SmoothDiff directly addresses the SGP, as we demonstrate in Appendix H. We further discuss in Appendix J how SmoothDiff's expected network Jacobian increases the *rank* compared to the regular network Jacobian $\boldsymbol{J}_f$, a desirable property for explanation methods that has been found to allow for more expressiveness and higher variation of explanations across classes [29].

## 2.5 Limitations

While SmoothDiff can be applied to arbitrary models, performance benefits are obtained by deriving performant VEJP (see Section 2.3). Such VEJPs have yet to be derived for attention layers, currently negating SmoothDiff's performance benefits over SG on ViT [32]. This limitation could be addressed through partial application of SmoothDiff, falling back to VJPs when no VEJPs are available.

## 3 Experiments

We evaluate SmoothDiff and SG on a pre-trained VGG-19 model [14] and the ImageNet dataset [33]. All experiments were run on a *NVIDIA A100 80GB* GPU and *AMD EPYC 9124* CPU. We use the *Julia-XAI* ecosystem [34, 35], *Flux.jl* deep learning framework [36, 37] and implement VEJPs using *ChainRules.jl* [38] and *Zygote.jl* [39].

We provide the complete source code to reproduce our experiments at `https://github.com/adrhill/smoothdiff-experiments`, including SmoothDiff reference implementations in Julia [40] and PyTorch.

---

[2]The suggested sample size for SG [6].

[3]SmoothDiff requires $n + k = 1050$ forward and backward passes, whereas SG requires $2nk = 10^5$.

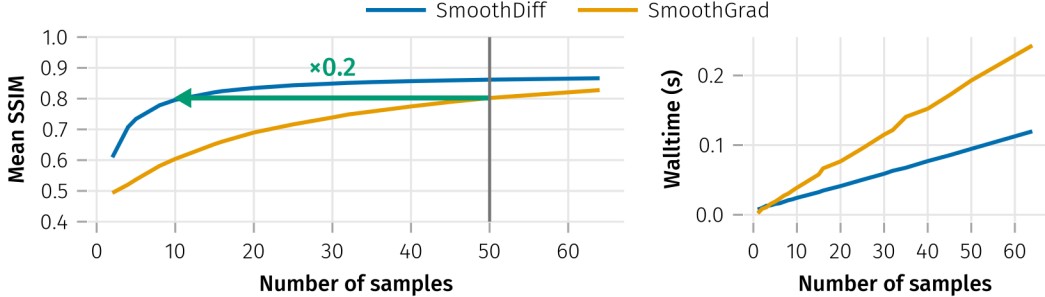

Figure 2: SmoothDiff improves on SG in terms of both *sample efficiency* and *computational performance*. **Left**: Improvements in sample efficiency, measured by the similarity to converged SG explanations in terms of SSIM [41]. At the recommended sample size of 50, SG achieves a SSIM of 0.8. SmoothDiff requires five times fewer samples to achieve a similar SSIM, highlighted by the green arrow. **Right:** Improvements in computational efficiency, measured by walltime per sample. Both SmoothDiff and SG scale linearly, however SmoothDiff is approximately twice as fast across sample sizes.

### 3.1 Convergence comparison

Since the Gaussian convolution (1) has no analytical solution, we use converged[4] SG explanations over a batch of 128 randomly drawn ImageNet [33] images as our reference. Convergence towards this reference is quantified using the *structural similarity index measure* (SSIM) [41]. A standard deviation of $\sigma = 0.5$ is used for the sampling distribution $\mathcal{N}_g$ in both SmoothDiff and SG.

The resulting mean SSIM are shown on the left of Figure 2. Since SG is used as a reference, it by construction approaches a SSIM of 1 with increasing sample counts. We observe that through the factorization in (5), SmoothDiff trades a small inherent bias in form of neglected cross-covariance terms for a large increase in sample efficiency. Using the recommended sample size of $n = 50$, SG reaches a SSIM of 0.8, whereas with SmoothDiff, the same SSIM is obtained in just 10 samples, resulting in increased sample efficiency of factor 5. For a discussion of the neglected cross-covariance terms, refer to Appendix E.

### 3.2 Benchmarks

We benchmark the performance of SmoothDiff and SG as a function of sample size $n$ by computing explanations for batches of 128 ImageNet images. As shown on the right side of Figure 2, SmoothDiff is approximately twice as fast as SG across all sample sizes, further compounding the increased sample efficiency demonstrated in Section 3.1. These computational performance improvements match theoretical considerations in Section 2.4, where single-class explanations using SmoothDiff are shown to require $n + 1$ forward and backward passes, whereas SG requires $2n$.

### 3.3 Heatmaps

For a qualitative comparison of SmoothDiff and SG convergence behavior we visualize heatmaps over increasing sample sizes in Figure 3. Even at small sample sizes, SmoothDiff leads to well-contoured heatmaps with little noise, while SG only progressively reduces noise effects as a function of $n$; it becomes apparent from Figure 3 that even at the recommended parametrization of SG, i.e. $n = 50$, there is further room for improvement. Further heatmaps including examples of multi-class explanations can be found Appendix K.

### 3.4 Quantitative results

In addition to the qualitative results in Section 3.3, we use pixel-flipping to quantify SmoothDiff's performance. In the absence of a ground truth, pixel-flipping measures the *faithfulness* of an XAI method by perturbing features in the order of their attribution strength. We use the *symmetric*

---

[4]More specifically, SmoothGrad explanations using $n = 10^6$ samples.

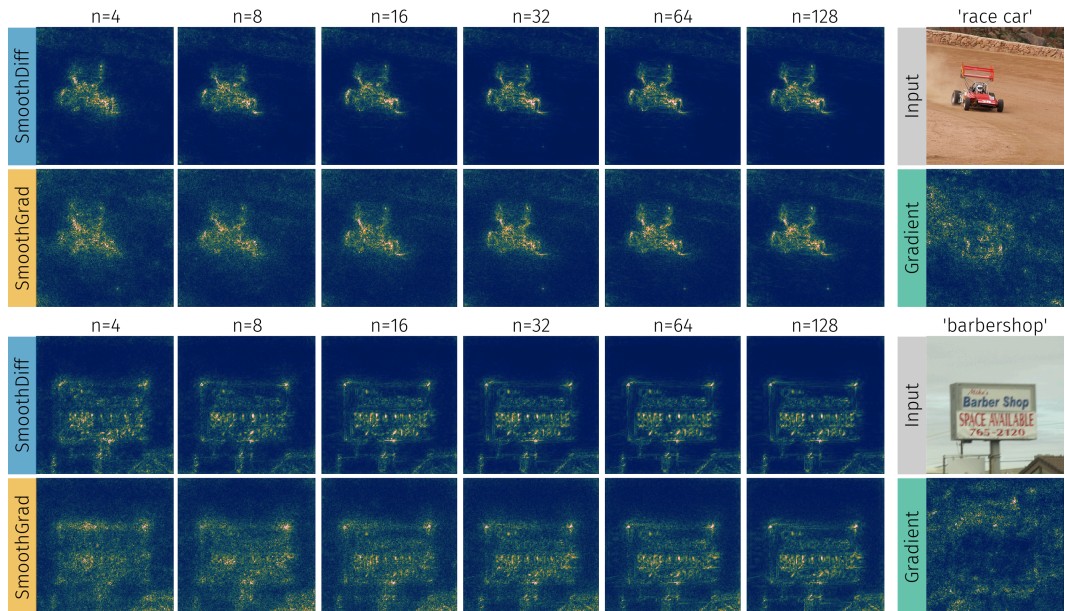

Figure 3: Qualitative comparison of SmoothDiff and SG heatmap convergence over increasing sample sizes. Even at very small sample sizes, SmoothDiff leads to well-contoured heatmaps with little noise.

*relevance gain* (SRG) measure [42], which largely removes the influence of imputing methods on pixel-flipping. This is achieved by computing the area under two pixel-flipping curves: one deleting features in order of *most influential first* (MIF), the other in order of *least influential first* (LIF).[5] The SRG is defined as the difference between the LIF and MIF measures. Figure 4 shows the results using mean imputing and a standard deviation of $\sigma = 0.5$ for SmoothDiff and SG. SmoothDiff reaches a higher peak SRG and does so at a lower sample size. It remains superior or equal to SmoothGrad up to approximately the recommended sample size of SmoothGrad.

We additionally compare optimal SmoothDiff and SmoothGrad settings from Figure 4 to regular gradient explanations and $\beta$-Smoothing (see Section 1.3 and Appendix G), which also attempts to smooth gradients. The pixel-flipping results are summarized in Table 1, demonstrating that SmoothDiff outperforms other methods in the SRG, as well as the LIF and MIF measures [42].

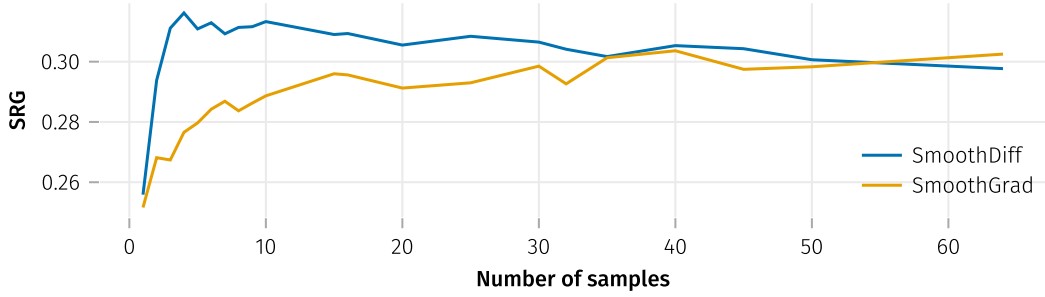

Figure 4: Comparison of SmoothDiff and SG in terms of pixel-flipping performance, measured by SRG [42]. While the performance of SG increases with sample size up to $n = 40$, SmoothDiff's performance peaks at just $n = 4$ samples.

---

[5]In terms of the area under the pixel-flipping curve, deleting LIF is equivalent to inserting MIF. The SRG measure therefore combines insertion and deletion tests.

Table 1: Pixel-flipping results (↑: higher is better, ↓: lower is better).

| Method | Parameters | LIF ↑ | MIF ↓ | SRG ↑ |
|---|---|---|---|---|
| SmoothDiff | $\sigma = 0.5, n = 4$ | **0.382** | **0.066** | **0.316** |
| SmoothGrad | $\sigma = 0.5, n = 40$ | 0.373 | 0.069 | 0.304 |
| $\beta$-Smoothing | $\beta = 1.0$ | 0.316 | 0.086 | 0.230 |
| $\beta$-Smoothing | $\beta = 2.0$ | 0.311 | 0.093 | 0.218 |
| $\beta$-Smoothing | $\beta = 0.5$ | 0.293 | 0.098 | 0.195 |
| Gradient | — | 0.276 | 0.111 | 0.166 |

For additional quantitative results in terms of *localization*, *complexity* and *robustness*, refer to Appendix F, which compares SmoothDiff with gradients, SG, *SmootGrad-Squared* [43], *Integrated Gradients* [44], *LRP composites* [45, 46], *GradCAM* [47], and several random baselines.

## 4   Discussion

SmoothGrad is a popular gradient-based choice for explaining ML models based on convolving gradients. While originally conceived for compensating shattered gradient effects, it is challenged by the need to accurately approximate the convolution, which would, in principle, require expensive sampling. In practice, SmoothGrad users compromise by trading accuracy with speed. In our work we contributed by proposing SmoothDiff which is making use of the well-known mathematical structure of the Jacobian in neural networks together with automatic differentiation functionalities. Specifically, we show that the expected Jacobian can be decomposed efficiently into independent parts, with a focus on the relevant network non-linearities only. Automatic differentiation helps to readily implement this idea using vector-Jacobian products without further work. This results in a higher accuracy of the smoothed gradient convolution estimation at fewer samples: resulting in speedups of up to two orders of magnitude for multi-class explanations over the original Smooth-Grad. This means that SmoothDiff helps to enhance the usability (speed and quality) of SmoothGrad greatly – as seen across all simulations and benchmarks explored. As smoothing comes with a reduction of curvature or second order derivatives, we conjecture that smoothed explanations as the ones studied here may also provide a higher resilience in the spirit of [23].

We can see three potentially interesting directions to further improve SmoothDiff: (i) Instead of approximating Gaussian convolutions by random samples, more efficient means of quadrature should be investigated. (ii) Moving beyond Gaussian filtering over the entire input domain, the geometry of the data manifold can be taken into account for convolutional smoothing, as shown in [48]. Insights from SmoothDiff could also be applied to other methods integrating over sensitivities, e.g. *Integrated Gradients* [44]. (iii) Instead of sampling activations on multiple forward passes, probability distributions could be propagated in a single forward pass, possibly further accelerating the method.

Future work will furthermore explore smoothed differentiation as in SmoothDiff beyond XAI, for example in applications of AI for the sciences where smooth priors in terms of differential equations exist, say, in the laws of physics.

## Contributions

AH devised the project and main conceptual ideas. He wrote the majority of the manuscript, including experiments, benchmarks, figures, tables and code. NM contributed to the initial design of the method during his lab rotation under the guidance of AH. He evaluated the 1D example in Figure 1(a) and Appendix B, compared color schemes in Appendix D, wrote the initial draft of Appendix H, created the ReLU pseudocode listing in Appendix I, and plotted Figure 2 from AH's experimental results. JM wrote the PyTorch implementation of SmoothDiff, evaluated the *GridPG* experiments in Appendix F.2 and contributed to Section 2.4. SB advised the pixel-flipping experi-

ments in Section 3.4 and helped frame the paper. KRM supervised the project and helped frame, structure and write the paper. All authors discussed the results and commented on the manuscript.

## Acknowledgements

We would like to thank the anonymous reviewers for their feedback, especially reviewer "Meht". We also thank Christopher Anders, Shinichi Nakajima, Anna Hedström, and Stefan Gugler for their valuable feedback and insightful discussions. Furthermore, we gratefully acknowledge funding from the German Federal Ministry of Education and Research under the grant BIFOLD25B. KRM was partly supported by the Institute of Information & Communications Technology Planning & Evaluation (IITP) grant funded by the Korea government (MSIT) (No. RS-2019-II190079, Artificial Intelligence Graduate School Program, Korea University) and grant funded by the Korea government (MSIT) (No. RS-2024-00457882, AI Research Hub Project).

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

## A    Notation

We denote scalar quantities by lowercase letters $x$, vector quantities by bold lowercase letters $\boldsymbol{x}$ and matrices by bold uppercase letters $\boldsymbol{X}$. The coefficients in a vector $\boldsymbol{x}$ are written as $x_i$, the rows of a matrix $\boldsymbol{X}$ as $\boldsymbol{X}_{i,:}$ and the coefficients as $[\boldsymbol{X}]_{i,j} = X_{i,j}$, using row and column indices respectively. All vectors $\boldsymbol{x}$ are column-vectors and all $\boldsymbol{x}^T$ are row-vectors. For a vector-to-scalar function $f : \mathbb{R}^n \to \mathbb{R}$, we use $\nabla f(\boldsymbol{x}) \in \mathbb{R}^n$ for the gradient vector. For a vector-to-vector function $f : \mathbb{R}^n \to \mathbb{R}^m$, we denote as $\boldsymbol{J}_f(\boldsymbol{x}) \in \mathbb{R}^{n \times m}$ the Jacobian matrix of first-order partial derivatives $\frac{\partial f_i(\boldsymbol{x})}{\partial x_j}$. The $i$-th basis vector is written as $\boldsymbol{e}_i$. We reserve the index $c$ for the selection of the output neurons, such that for a neural network $f$ and an input $\boldsymbol{x}$, the predicted logit for the $c$-th class corresponds to $f_c(\boldsymbol{x})$.

## B    Experiment details – Figure 1

**(a) 1-dimensional toy example:** To demonstrate the SGP on gradient-based XAI methods in Figure 1 (a), we define a deep neural network similar to [13]. We use a sequence of 16 fully connected ReLU layers with 64 hidden neurons each, mapping scalar inputs to scalar outputs. Both weights and biases are drawn from a normal distribution with mean zero and variance $\frac{1}{N}$. Both SmoothDiff and SG explanations are computed using $\sigma = 0.05$ and $n = 10$ samples. Since the explanations are stochastic (inputs are randomly sampled), we visualize the means and 5th- to 95th-percentile bands over 1024 evaluations per input. $\beta$-Smoothing was computed using $\beta = \log(2)\sqrt{\frac{2\pi}{\sigma}}L \approx 117$, motivated by scaling up the analytical derivation for a single layer [23]. The Gaussian convolution was computed numerically from the gradient.

**(b) Vision model:** Explanations are computed on a pretrained VGG-19 model [14] and randomly selected images from the ImageNet dataset [33]. They are visualized using the heatmapping techniques described in Appendix D. Both SmoothDiff and SG are computed using $\sigma = 0.5$. Since the analytical Gaussian convolution is intractable and an accurate numerical computation is infeasible, it was approximated by computing SG with $n = 10^6$ samples (see Section 3.1).

## C    Relevance pooling

Most XAI methods, including SmoothDiff, have in common that the dimensionality of their numerical explanations matches that of the input features. The inputs to ImageNet models are (normalized) RGB images: three-dimensional tensors that include width, height and color channel dimensions. Explanations of these inputs therefore also contain three separate values per pixel, one for each color channel (red, green, blue). For the purpose of heatmapping (see Appendix D) as well as several evaluation metrics (see Appendix F), these three values need to be reduced to a single scalar value by applying a *relevance pooling* function $R_{\mathrm{pool}} : \mathbb{R}^3 \to \mathbb{R}$ to each pixel [49]. Popular options include summation, max pooling and the $\ell^2$-norm (hereafter referred to as norm-pooling).

As motivated in Section 1, explanations can be based on different mathematical objects, e.g. gradients, Taylor expansions and Shapley values. The choice of relevance pooling depends on the XAI method: For LRP, summation is the most common choice, as it maintains its property of *relevance conservation* [7]. Sum-pooled explanations can contain both positive and negative values and are therefore commonly visualized using diverging color maps, see e.g. LRP's characteristic red-white-blue heatmaps.

For gradient-based methods however, summation is not an option. We want to illustrate this with a thought experiment: When using a gradient-based method to explain why an image of a red fire truck is correctly classified by an idealized robust model, we might expect the gradient to be positive on the red color channel (a more intense red positively affecting the classification) and negative on the blue and green color channels (also intensifying the red color). In this example, negative sensitivities on the blue and green color channels do not imply that the pixel is of negative importance to the classifier. Sum-pooling should therefore be avoided, as it can in the worst case result in a negative value even though the pixel was of high sensitivity. Instead, we suggest the use of the norm-pooling, measuring the pixel-wise gradient magnitude. Since norm-pooled explanations are non-negative,

they should be visualized with sequential color maps (see Appendix D). As we will demonstrate in Appendix F, the choice of relevance pooling has a significant impact on the evaluation of XAI methods, a crucial aspect that remains underexplored in the literature.

# D Heatmapping

We visualize numerical explanations as heatmaps using the *VisionHeatmaps.jl* package [34] from the *Julia-XAI* ecosystem [35]. To map gradient-based explanations onto a color scheme, we reduce the color channel dimension by applying norm-pooling (see Appendix C). Finally, to avoid desaturation due to numerical outliers, the color scheme is mapped to a range between 0 and the 99.9th percentile of the pooled explanation, effectively normalizing values within the explanation.

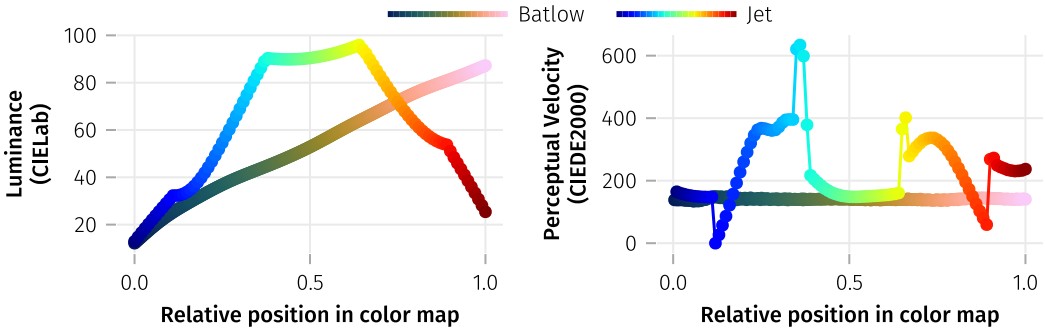

Figure 5: Comparison of perceptual properties of the `jet` and `batlow` color maps.

We use the perceptually uniform `batlow` color map [50, 51]. For sequential color maps, a linear increase in luminance is desirable, as it is the primary perceptual dimension perceived as intensity, as well as the one best resolved at high frequencies [52]. Additionally, constant change in overall perceptual difference between adjacent colors ensures uniformity [53]. Figure 5 compares `batlow` with the popular `jet` color map, measuring luminance $L^*$ in the *CIELab* color space and overall perceptual velocity using the *CIEDE2000* distance metric [54]. While we discourage the use of the `jet` color map due to its perceptual non-uniformity, especially in combination with highly processed transparent overlays, we provide such SmoothDiff heatmaps in Figure 6 due to their ongoing popularity in the field.

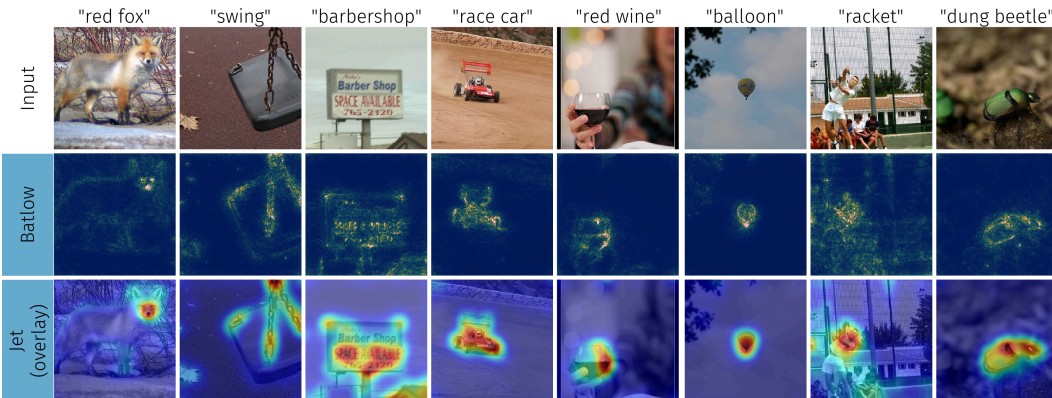

Figure 6: Comparison of heatmapping techniques on SmoothDiff explanations ($n = 10$, $\sigma = 0.5$).

# E    Characterizing neglected cross-covariances

As discussed in Section 3.1, SmoothDiff trades a small inherent bias in form of neglected cross-covariance terms in (5) for a large increase in sample efficiency. To quantify the bias, we modify the convergence experiment from Section 3.1, treating SmoothDiff and SG explanations (i.e. smoothed gradients) as vectors. We then measure their similarity to the converged SG reference explanation by computing cosine similarities and magnitudes.

The resulting convergence plots are shown Figure 7: In terms of cosine similarity, SmoothDiff shows superior convergence behavior up to SG's recommended sample size of $n = 50$. This indicates that the direction of SmoothDiff explanations aligns well with the reference. In terms of magnitude however, SmoothDiff has a clear bias: While SG converges towards the magnitude of the reference (i.e. a relative magnitude of one), SmoothDiff converges towards a much smaller relative magnitude of 0.461. We suspect that this difference in magnitude is caused by the neglected cross-covariances.

As described in Appendix D, explanations are relevance pooled and normalized during heatmapping. Heatmaps therefore only depend on the direction of the explanation vector, not the magnitude, limiting the effect of neglected cross-covariances. This observation is in line with the SSIM convergence plot in Figure 2.

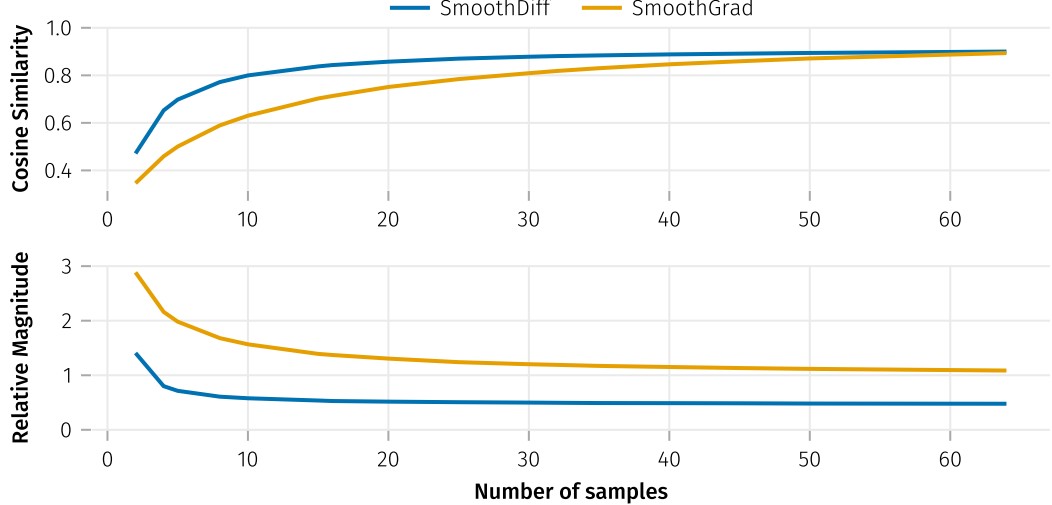

Figure 7: Convergence behavior of SmoothDiff and SG towards the reference explanation in terms of cosine similarity (upper) and relative magnitude of the explanation (lower). Convergence with the reference is obtained at a cosine similarity and relative magnitude of one.

# F    Further quantitative results

In addition to the pixel-flipping experiments which evaluate *faithfulness* in Section 3.4, we also evaluate metrics in the domains of *robustness*, *localization* and *complexity* using the *Quantus* toolkit [55]. Since some localization metrics require ground truth segmentation masks that localize the object of the target class, a dataset of 256 input images, labels and segmentation masks was created by sub-sampling correctly classified images from the ImageNet-S-50 dataset [56].

In addition to the Gradient, SmoothDiff and SG methods in Section 3, we also evaluate *SmoothGrad-Squared* [43], *Integrated Gradients* [44], *GradCAM* [47], two *LRP composites* [45, 46], and three random baselines. Reference implementations from the *Captum* software package [57] are used for Integrated Gradients and GradCAM, while *Zennit* [58] is used for LRP, namely the `EpsilonPlus` and `EpsilonAlpha2Beta1` composites. Metrics are evaluated on pre-trained VGG-19 and ResNet-18 models using Quantus' default parameters.

As motivated in Appendix C, relevance pooling significantly impacts both heatmapping and evaluation metrics. Since the goal of XAI is to further humans' understanding of machine learning models,

we argue that evaluation metrics should apply the same relevance pooling functions that are used to compute heatmapping visualizations. We therefore evaluate gradient-based methods (Gradient, SmoothGrad, SmoothDiff, Integrated Gradients) with norm-pooling and methods based on Taylor expansions (LRP composites, Input x Gradient) with sum-pooling.

In [43], a variant of SG called SmoothGrad-Squared (SG-SQ) is shown to perform well using sum-pooling. Instead of averaging gradients, SG-SQ averages squared gradients.[6] While not exactly analogous,[7] we introduce SmoothDiff-Squared (SD-SQ), which we define as the square of a SmoothDiff explanation

$$[\text{SD-SQ}(\boldsymbol{x})]_i = [\text{SD}(\boldsymbol{x})]_i^2 \ . \tag{10}$$

We want to emphasize that a sum-pooled SD-SQ explanation is the square of a norm-pooled SmoothDiff explanation, blurring the lines between XAI method and relevance pooling. To further quantify the influence of relevance pooling on evaluation metrics, we introduce three random baselines that are equally uninformative:

- **Random ($\ell^2$):** random explanation drawn from a normal distribution, norm-pooled
- **Random ($\Sigma$):** random explanation drawn from a normal distribution, sum-pooled
- **Random-Squared ($\Sigma$):** squared random explanation drawn from a normal distribution, sum-pooled

For SmoothDiff, SmoothDiff-Squared, SmoothGrad, SmoothGrad-Squared, and Integrated Gradients, explanations are computed using 20 samples in an attempt to make them computationally comparable. All evaluation metrics are evaluated on batches of 256 images described above. We visualize the resulting distributions of scores as box plots in Figures 8 to 13, additionally reporting means in the right-hand column. XAI methods and random baselines are purposefully arranged in three groups: gradient-based methods using norm-pooling, squared methods with sum-pooling (SG-SQ, SD-SQ, Random-Squared) and other sum-pooled methods.

### F.1 Robustness

Robustness metrics evaluate how stable explanations are under slight input perturbations. We evaluate two robustness metrics, the first being the *Local Lipschitz Estimate* (LLE), described in [59, 60]. The resulting distribution of scores is visualized in Figure 8, where lower scores indicate higher robustness. On VGG-19, SmoothDiff-Squared outperforms all other methods by a significant margin. Results on ResNet-18 qualitatively mirror those of VGG-19, indicating that SmoothDiff and SmoothDiff-Squared explanations are robust across model architectures. LRP `EpsilonAlpha2Beta1` does however significantly improve on ResNet-18, achieving a slightly lower mean than SmoothDiff-Squared. Within gradient-based methods, SmoothDiff is the most robust on both models. The three random baselines reveal that the LLE is biased towards Random-Squared,[8] mirroring the fact that SmoothDiff-Squared outperforms SmoothDiff.

The second evaluated robustness metric is *Average Sensitivity* [61], shown in Figure 9. Once again, lower scores indicate higher robutstness. SmoothDiff's robustness on both VGG-19 and ResNet-18 is only outperformed by the two LRP composites. Within gradient-based methods, SmoothDiff again is the most robust method across model architectures. The random baselines reveal that the metric is biased towards norm-pooling, mirroring the fact that SmoothDiff outperforms SmoothDiff-Squared.

### F.2 Localization

Localization metrics test whether the highest values of an explanation is located within a desired region of interest. In case of the first metric, *Relevance Rank Accuracy* (RRA) [49], the region of interest is specified by providing ground-truth segmentation masks, as described above. The resulting RRA scores are shown in Figure 10, where higher scores are desirable. In the three random baselines,

---

[6]When talking about squared gradients and explanations, we refer to the element-wise square.

[7]In terms of (1), SG-SQ approximates the expected value of the squared gradients while SD-SQ approximates the square of the expected value of gradients.

[8]Since all random baselines are equally informative, we would expect them to be equally robust.

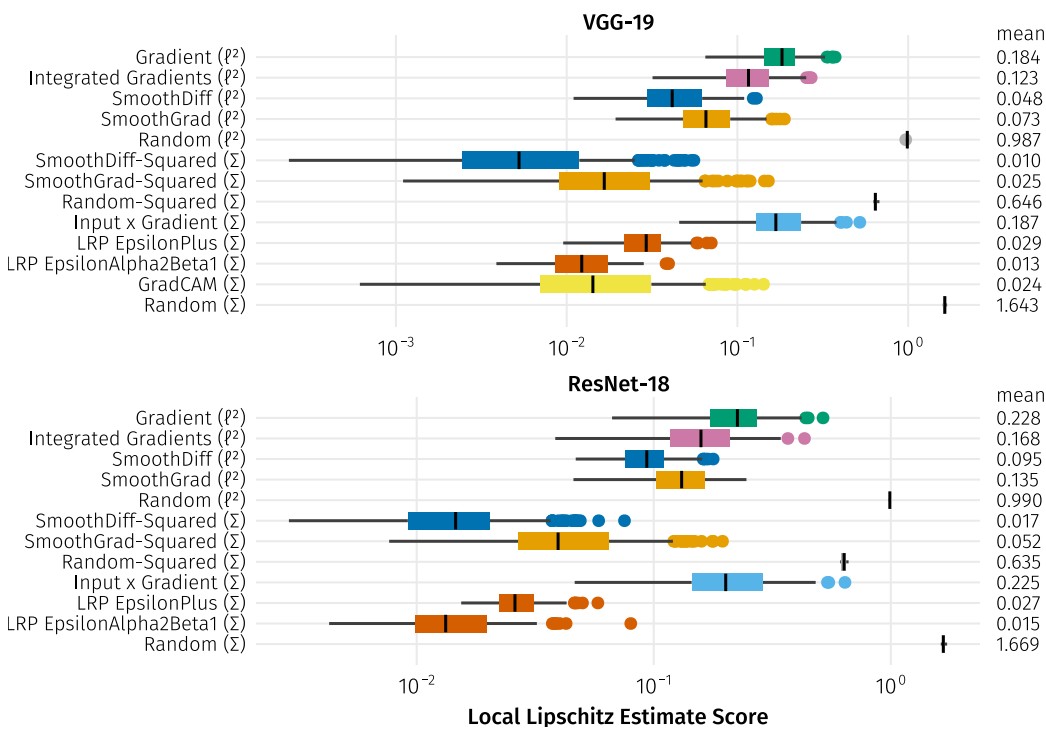

Figure 8: Evaluation of the *Local Lipschitz Estimate* [59, 60] as a measure of robustness. Lower scores indicate a higher robustness. We indicate relevance pooling functions next to XAI method names, using $\ell^2$ for norm-pooling and $\Sigma$ for sum-pooling.

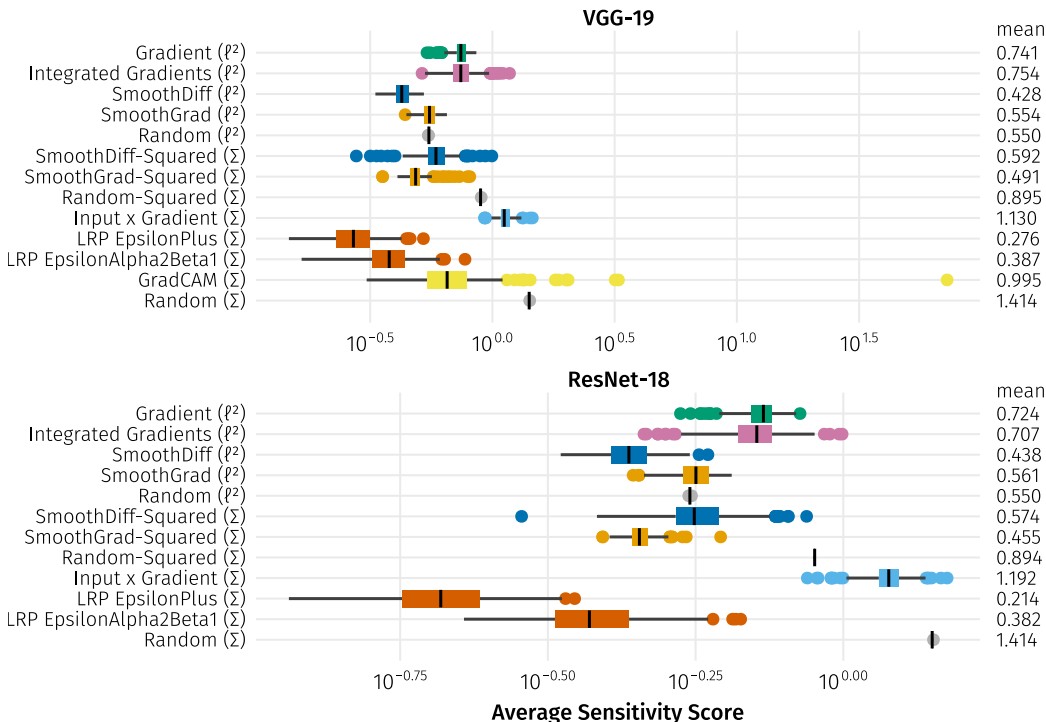

Figure 9: Evaluation of *Average Sensitivity* [61] as a measure of robustness. Lower scores indicate a higher robustness. We indicate relevance pooling functions next to XAI method names, using $\ell^2$ for norm-pooling and $\Sigma$ for sum-pooling.

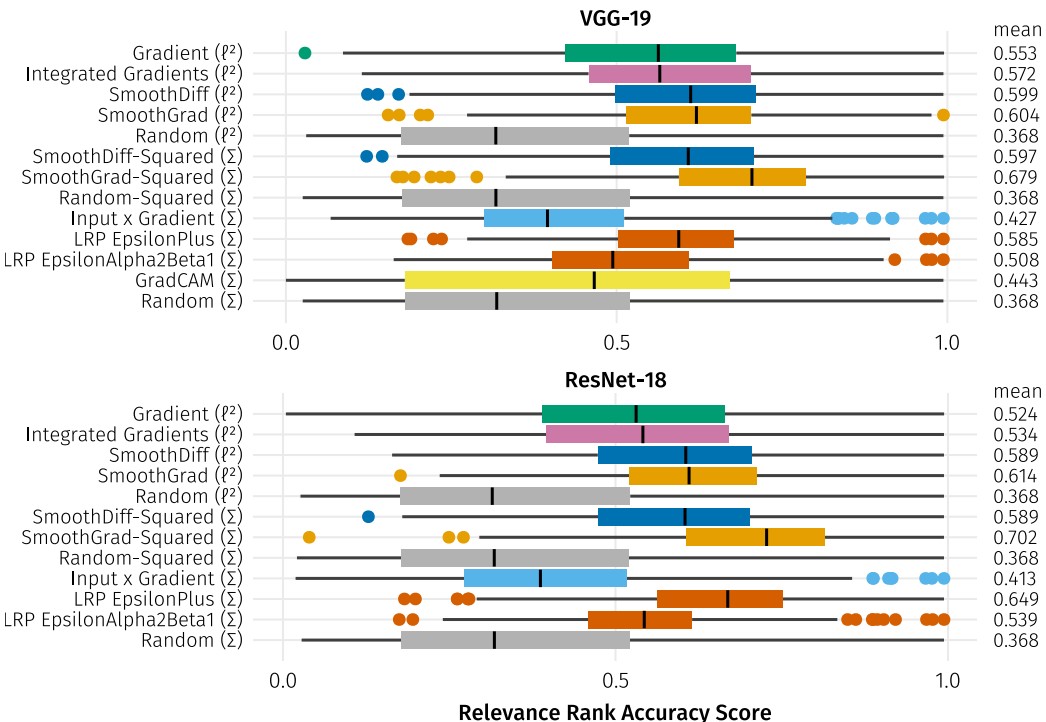

Figure 10: Evaluation the *Relevance Rank Accuracy* [49] as a measure of localization. Higher scores indicate better localization. We indicate relevance pooling functions next to XAI method names, using $\ell^2$ for norm-pooling and $\Sigma$ for sum-pooling.

the RRA shows no visible bias towards any of the evaluated relevance-pooling functions. SG-SQ performs best on both VGG-19 and ResNet-18, followed by SG, SmoothDiff and SmoothDiff-Squared, which all perform similarly well.

The second localization metric is the *Grid Pointing Game* (GridPG) [62, 63], an artificial benchmark in which four images from different classes are tiled in a $2 \times 2$ grid. The metric selects one of the images as the region of interest and target class of the explanation. Scores correspond to the sum of positive values within the target quadrant, divided by the sum of all positive values, normalizing scores between zero and one. The results are shown in Figure 11, where higher scores indicate better localization. Since explanations from all three random baselines are distributed equally across all four quadrants, all baselines obtain a score of 0.25. Due to its normalization, the GridPG appears to favor XAI methods (and corresponding relevance pooling functions) like LRP that can return negative values.[9] Since this is neither the case for gradient-based methods, nor for squared methods, these perform significantly worse than LRP composites and GradCAM.

### F.3 Complexity

The final category of evaluation metrics quantifies the conciseness of explanations, favoring highlighting few features. The first metric is *Complexity* [64], shown in Figure 12. On VGG-19, GradCAM performs best, followed by SmoothDiff-Squared. On ResNet-18, SmoothDiff-Squared performs best (in terms of mean score) as Captum's GradCAM implementation could not be evaluated on the model. The random baselines reveal a bias for Random-Squared and the sum-pooled random explanation, mirroring the fact that SmoothDiff-Squared outperforms SmoothDiff.

The second and final complexity metric is *Sparseness* [65], which is based on the Gini Index. Results are shown in Figure 13. On VGG-19, they qualitatively mirror those of the complexity metric in Figure 12: GradCAM performs best, followed by SmoothDiff-Squared and LRP `EpsilonAlpha2Beta1`. On ResNet-18, SmoothDiff-Squared is a close second behind the LRP `EpsilonAlpha2Beta1`.

---

[9]Non-negative explanations will result in a larger denominator.

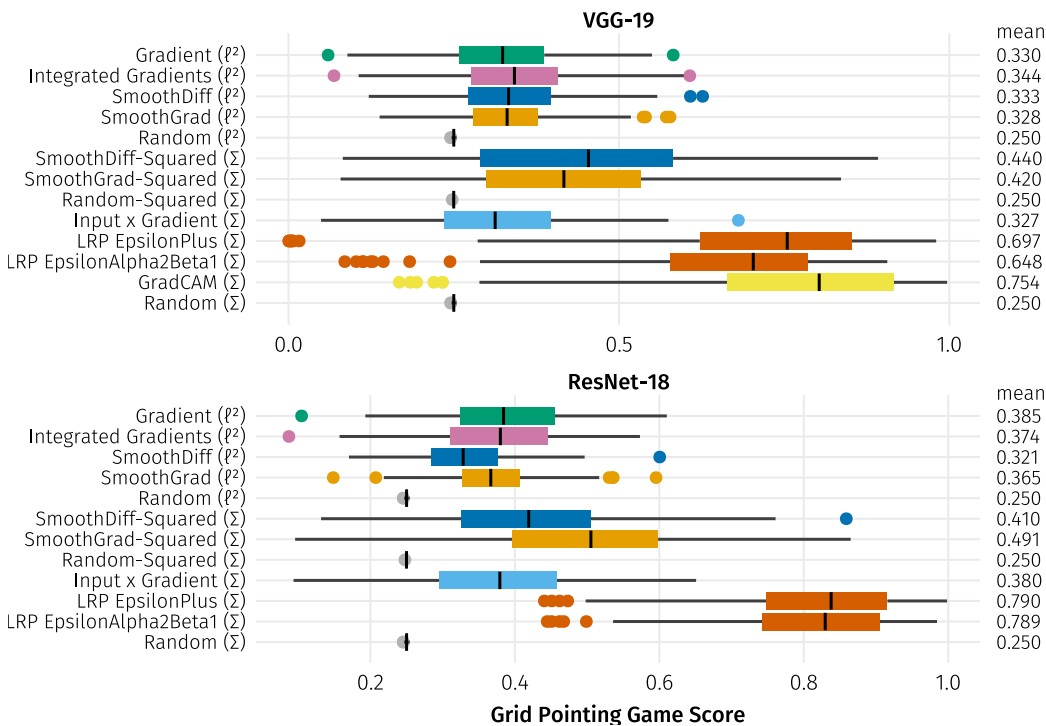

Figure 11: Evaluation of the *Grid Pointing Game* [62, 63] as a measure of localization. Higher scores indicate better localization. We indicate relevance pooling functions next to XAI method names, using $\ell^2$ for norm-pooling and $\Sigma$ for sum-pooling.

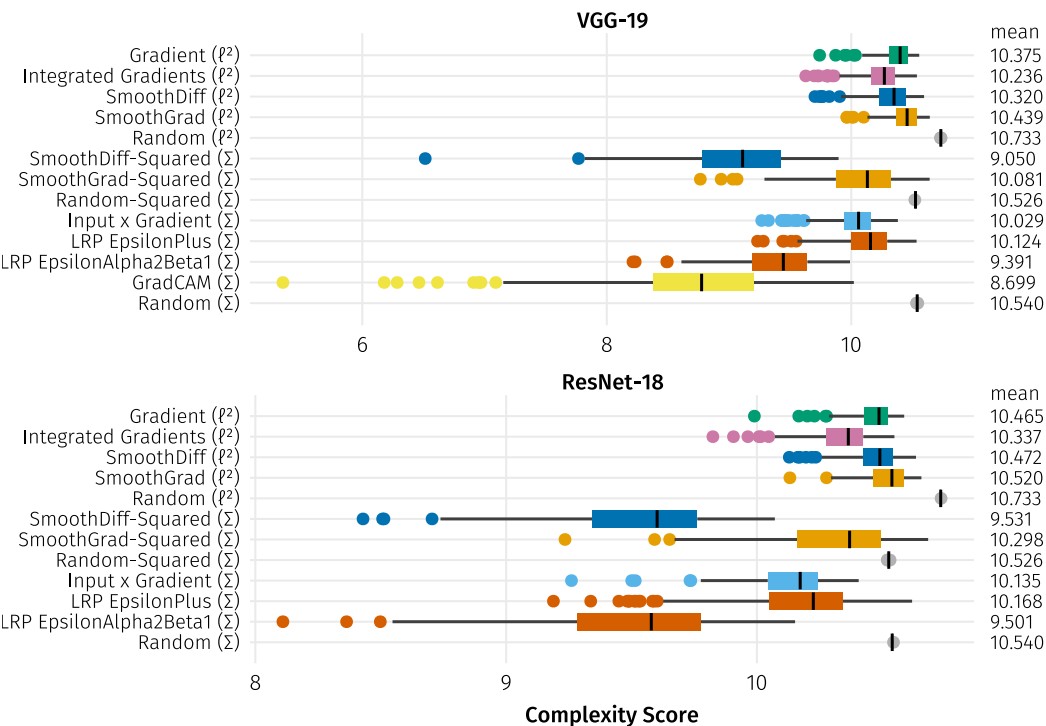

Figure 12: Evaluation of *Complexity* [64]. Lower scores indicate a better complexity. We indicate relevance pooling functions next to XAI method names, using $\ell^2$ for norm-pooling and $\Sigma$ for sum-pooling.

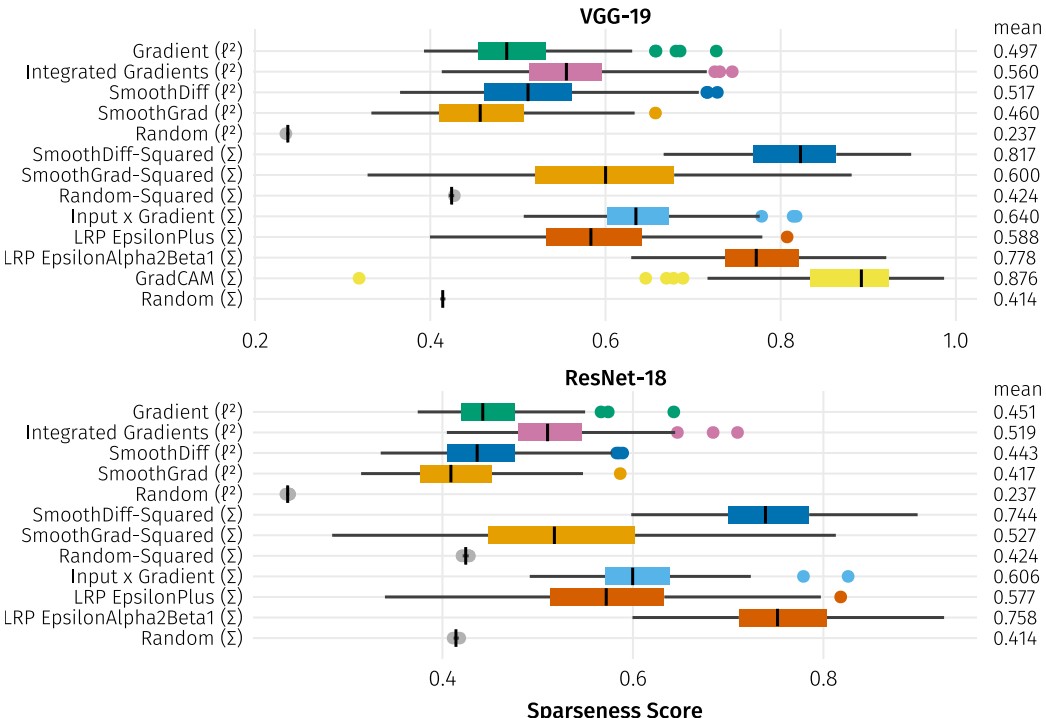

Figure 13: Evaluation of *Sparseness* [65] as a measure of complexity. Higher scores indicate a better complexity. We indicate relevance pooling functions next to XAI method names, using $\ell^2$ for norm-pooling and $\Sigma$ for sum-pooling.

The metric shows a significant bias against the norm-pooled random explanation, which can also be seen in the comparatively low scores of all gradient-based explanations.

In conclusion, SmoothDiff and SmoothDiff-Squared perform very well in terms of localization, robustness and complexity, three desired properties in XAI, while being performant and simple to implement. Using three equally uninformative random baselines (which should have performed equally well in terms of robustness, localization and complexity), we demonstrated that relevance pooling functions strongly alter the scores of most metrics, highlighting their significant (but underexplored) impact on the evaluation and development of XAI methods, as well as the resulting difficulty of comparing methods that require different relevance pooling functions.

## G $\beta$-Smoothing and Gaussian convolutions

In [15], the analytical solution for the convolution of a ReLU activation function $r$ with a Gaussian kernel $g_\sigma$ is shown to be

$$(r * g_\sigma)(x) = \underbrace{\frac{1}{2}\left[1 + \mathrm{erf}\left(\frac{x}{\sqrt{2}\sigma}\right)\right]x}_{\Phi_\sigma(x)} + \sigma^2 g_\sigma(x) \tag{11}$$

with derivative $(r * g_\sigma)'(x) = \Phi_\sigma(x)$, where erf is the error function. The authors term these functions *smoothed ReLU* and *smoothed Heaviside* respectively. For a single-layer neural network, SmoothDiff's expected ReLU Jacobians (8) are the sampling approximation to the smoothed Heaviside function.

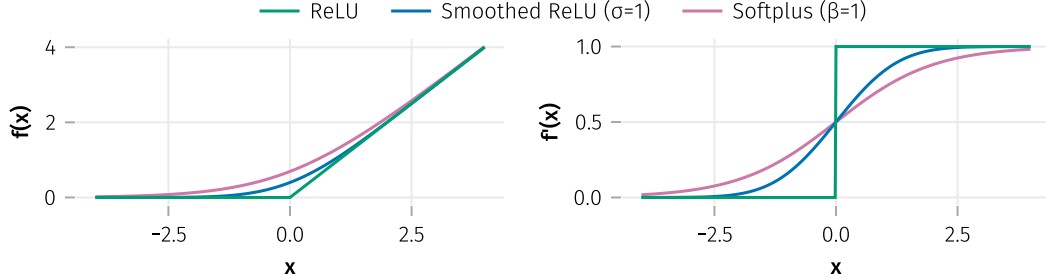

Figure 14: Comparison of ReLU, smoothed ReLU, and softplus functions (left) and their derivatives (right).

In [23], it is shown that gradient-based explanations on deep ReLU networks can be manipulated with hardly perceptible perturbations. The authors argue that this vulnerability is due to the large curvature of such networks and propose $\beta$-Smoothing as a robust explanation method, replacing ReLU activation functions by softplus activations

$$\text{softplus}_\beta(x) = \frac{1}{\beta} \log(1 + e^{\beta x}) \tag{12}$$

with small parameters $\beta$ around one (matching pixel-flipping results in Table 1, in which $\beta = 1$ performed best).

The authors demonstrate that for single-layer neural networks, $\beta$-Smoothing can be understood as an approximation to the infinite sample limit of SG and therefore an approximation to the Gaussian convolution (1). Figure 14 compares ReLU, smoothed ReLU and softplus functions, as well as their derivatives. $\beta$-Smoothing provides a good approximation of the smoothed ReLU and can be further improved by tuning the $\beta$ parameter.

While Gaussian convolutions are well approximated by $\beta$-Smoothing for single-layer networks, non-linear functions break the assumption of Gaussian activations in subsequent layers. Resulting heatmaps for $\beta$-Smoothing are shown in Figure 17. While the method qualitatively improves upon regular gradients, it results in heatmaps that approximate neither SG nor SmoothDiff and are more noisy and less contoured.

## H    SmoothDiff directly addresses the shattered gradient problem

Deep ReLU networks are piece-wise linear functions. The vector field of gradients of such networks is piece-wise constant and discontinuous, as shown in Figure 1 (a). In [66], it is shown that with increasing network depth, the number of linear regions and therefore discontinuities grows exponentially. These discontinuities are the source of the SGP for deep ReLU networks, which can be demonstrated using a path formulation of the gradient [13]. In Appendix H.1, we introduce our notation for a general path formulation similar to that of [67]. We apply it to deep ReLU networks in Appendix H.2, and demonstrate how SmoothDiff directly addresses it in Appendix H.3. We then extend the same formulation to general non-linearities in Appendix H.4.

### H.1    Notation

Given a neural network's directed computational graph, we denote by $P_{i \to c}$ the set of all paths $p$ from input $i$ to output $c$. Applying this notation to (3), the $i$-th entry of the gradient $\nabla f_c(\boldsymbol{x})$ can be written as

$$[\nabla f_c(\boldsymbol{x})]_i = \frac{\partial f_c(\boldsymbol{x})}{\partial x_i} = \sum_{p \in P_{i \to c}} \prod_n [\boldsymbol{J}_{f^n}(\boldsymbol{a}^n)]_p , \tag{13}$$

where $[\boldsymbol{J}_{f^n}]_p$ are scalar Jacobian entries of functions $f^n$ along the path.

We further factorize this product into:

- Linear or affine functions, whose Jacobian entries $\left[\boldsymbol{J}_{f^n}\right]_p = W_p^n$ are constant (for dense and convolutional layers, these correspond to individual weights).
- Non-linear functions, whose Jacobians entries $\left[\boldsymbol{J}_{f^n}(\boldsymbol{a}^n)\right]_p = g_p^n(\boldsymbol{a}^n)$ depend on one or more values in the input activation $\boldsymbol{a}^n$ (which in turn depend on $\boldsymbol{x}$, see Section 1).

This results in the path formulation

$$\frac{\partial f_c(\boldsymbol{x})}{\partial x_i} = \sum_{p \in P_{i \to c}} \underbrace{\prod_i W_p^i}_{=:W_p} \underbrace{\prod_j g_p^j(\boldsymbol{a}^j)}_{=:G_p(\boldsymbol{x})} = \sum_{p \in P_{i \to c}} W_p \cdot G_p(\boldsymbol{x}) \,. \tag{14}$$

Analogous to [13], we refer to the product $W_p$ as the *path-weight*. Note that since all Jacobian entries in (13) are scalar, products commute and the order of functions in the computational graph doesn't affect their contribution to $\partial f_c / \partial x_i$. Equation (14) therefore generalizes to both arbitrary compositional structures and arbitrary non-linearities.

## H.2 Deep ReLU networks

We now apply the path formulation of the gradient (14) to deep ReLU networks, which exclusively contain ReLU and max pooling non-linearities. For such networks, terms $g_p^j$ are indicator functions and evaluate to either one or zero (see Section 2.2). We denote a path $p$ as *active* if its product $G_p$ is non-zero. This is the case if and only if the path contains the largest inputs into every max pooling filter and all inputs to ReLUs on the path are positive. If a path is active, it contributes the path-weight $W_p$ to the gradient.

Since changes in the input $\boldsymbol{x}$ alter all activations $\boldsymbol{a}^j$, they also affect terms $g_p^j(\boldsymbol{a}^j)$. If even a single of these terms turns zero (e.g. because an activation into a ReLU becomes negative), contributions from the path-weight $W_p$ vanish. Reversely, changes in the input can also activate new paths, adding their respective path-weight $W_p$. With deeper networks, the length of paths increases linearly and their number exponentially, increasing the density of path activity changes over the input domain. As a result, small changes in the input cause more and more frequent changes in the gradient, eventually resembling white noise [13]. We can express this sensitivity of paths by the second-order partial derivative

$$\frac{\partial^2 f_c(\boldsymbol{x})}{\partial x_i \partial x_j} = \sum_{p \in P_{i \to c}} W_p \cdot \frac{\partial G_p(\boldsymbol{x})}{\partial x_j} \,. \tag{15}$$

Since $G_p$ are indicator functions, their derivatives at points of path activity change are infinite (or undefined). The sensitivity of paths in (15) directly depends on the curvature of $f_c$. Since $f_c$ is a piece-wise linear function, its curvature is either zero within a linear region or infinite (or undefined) at junction points of regions.

## H.3 Addressing shattered gradients

As described in Section 1, applying Gaussian convolutions (1) to piece-wise constant gradients $\nabla f_c(\boldsymbol{x})$ results in smooth continuous functions, introducing curvature (see Figure 1 a). Even though it doesn't directly act on the mechanisms described in Appendix H.2, SG therefore addresses the shattered gradient problem by approximating said Gaussian convolution in the infinite sample limit.

SmoothDiff addresses the issues in Appendix H.2 directly. By replacing Jacobians with expected Jacobians, SmoothDiff replaces indicator functions $g_p^n$ by smoothed, continuously differentiable functions $\widehat{g^n}_p = \left[\widehat{\boldsymbol{J}_{f^n}}\right]_p$. As a product of continuously differentiable functions, the resulting product $\hat{G}_p(\boldsymbol{x})$ is also continuously differentiable, avoiding the issue of path sensitivity (15) that characterize the SGP. Looking at sampling estimates more specifically (see Section 2.2), SmoothDiff activates all paths $p$ in which $g_p^n$ are non-zero in at least one sample. However, by ignoring covariances, it also

activates paths whose neurons are never jointly active in any individual sample and underestimates paths which are frequently active.

### H.4 General non-linearities

While Appendix H.2 is limited to deep ReLU networks, for which $G_p$ is an indicator function, the path sensitivity (15) also applies to general non-linearities. If a model contains non-linear functions with large second-order derivatives, the resulting terms $\partial G_p / \partial x_j$ can also become very large, directly increasing the sensitivity of path contributions and shattering the gradient. For a theoretical analysis of the influence of the curvature of $f_c$ on gradient shattering, refer to [68].

## I Example VEJP implementations

Algorithms 1 and 2 demonstrate the VEJP implementations for ReLUs and max pooling layers from Section 2.2. After replacing all `ReLU` activation functions in a model with `ReLU_SmoothDiff`, as well as replacing all `MaxPool` layers with corresponding `MaxPool_SmoothDiff`, $n$ forward passes of the model are computed with samples $\hat{x} \sim \mathcal{N}_g$, incrementing sample and activation counters. SmoothDiff can then be computed in a single `backward` pass through the model evaluating VEJPs $v^T \hat{J}$ (in the pseudocode, $v^T$ is named v and $\hat{J}$ is named J). For ReLUs, this computation corresponds to (9).

| | |
|---|---|
| 1 **class** ReLU_SmoothDiff | 1 **class** MaxPool_SmoothDiff |
| 2   ReLU | 2   MaxPool |
| 3   sample_count = 0 | 3   sample_count = 0 |
| 4   activation_count = 0 | 4   activation_count = 0 |
| 5   **function** forward(input) | 5   **function** forward(input) |
| 6     sample_count += 1 | 6     sample_count += 1 |
| 7     activation_count += input > 0 | 7     activation_count += is_max(input) |
| 8     **return** ReLU(input) | 8     **return** MaxPool(input) |
| 9   **function** backward(v) | 9   **function** backward(v) |
| 10     J = activation_count / sample_count | 10     J = activation_count / sample_count |
| 11     **return** v * J | 11     **return** v * J |

Algorithm 1: SmoothDiff VEJP for ReLUs.       Algorithm 2: SmoothDiff VEJP for max pooling layers.

## J Rank

For two matrices $A \in \mathbb{R}^{m \times k}$, $B \in \mathbb{R}^{k \times n}$, Sylvester's rank inequality tells us that

$$\operatorname{rank}(A) + \operatorname{rank}(B) - k \leq \operatorname{rank}(AB) \leq \min(\operatorname{rank}(A), \operatorname{rank}(B)). \qquad (16)$$

Using the factorization in (3), the rank of neural network Jacobians $J_f$ is therefore upper-bounded by the lowest-rank Jacobian $J_{f^n}$:

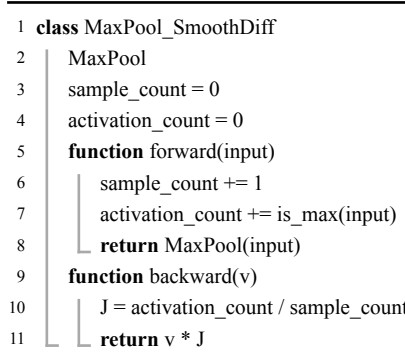

$$\operatorname{rank}(J_f) = \operatorname{rank}\left(\prod_n J_{f^n}\right) \leq \min_n \operatorname{rank}(J_{f^n}). \qquad (17)$$

In practice, the lowest-rank Jacobians of deep ReLU networks are those of element-wise ReLU activation functions (since weight matrices tend to have full rank). As discussed in Section 2.2, ReLU Jacobians are diagonal matrices with entries $J_{\sigma_{i,i}}(a) = \mathbb{1}_{x \geq 0}(a_i)$. Since the rank of a diagonal matrix corresponds to the number of non-zero entries, it is determined by the number of strictly positive activations $a_i$. SmoothDiff's expected Jacobian entries (8) are non-zero if a single sample leads to a positive activation $a_i$ (as discussed in Section 2.2 and Appendix H), directly increasing the rank of the Jacobian computed in (5).

## K    Sample heatmaps

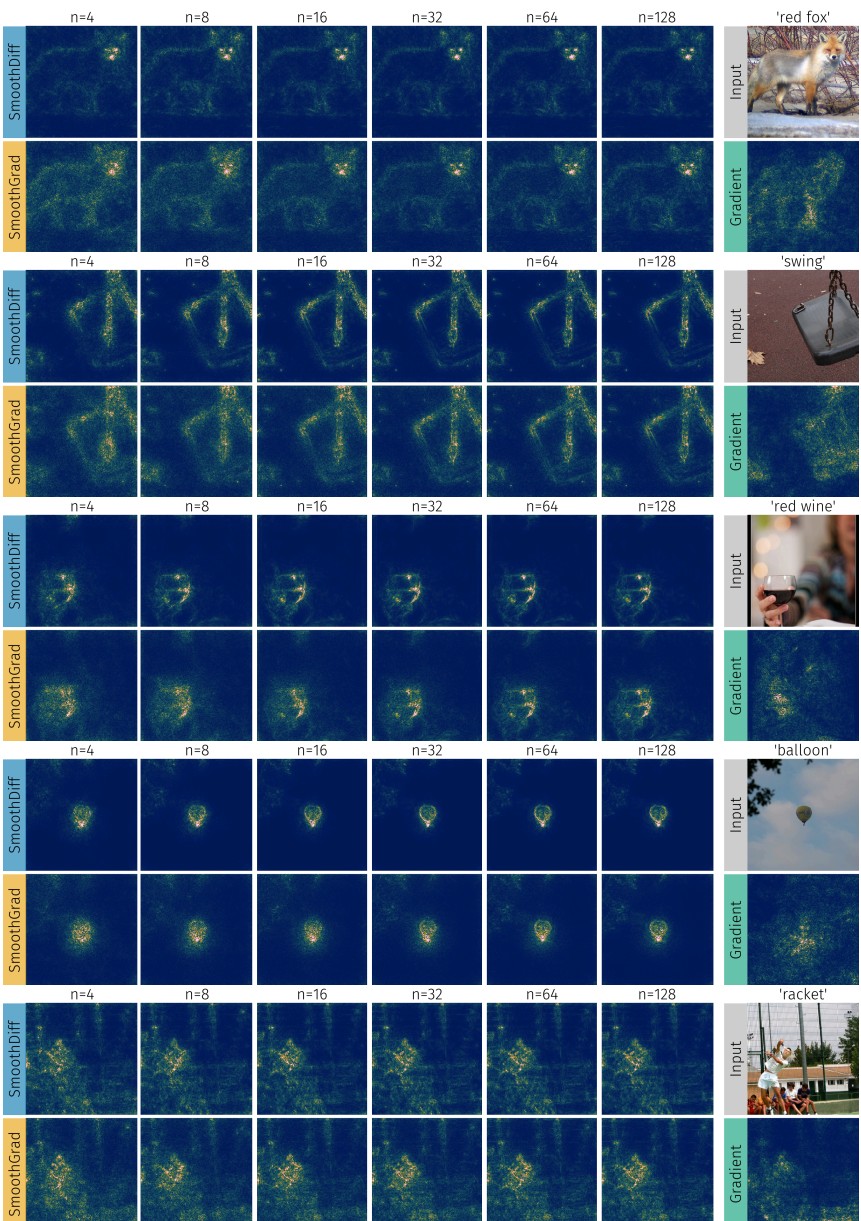

Figure 15: Qualitative comparison of SmoothDiff and SmoothGrad heatmap convergence over increasing sample sizes. Heatmaps are computed on VGG-19 using $\sigma = 0.5$.

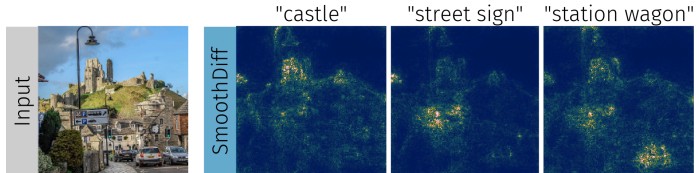

Figure 16: SmoothDiff multi-class explanations for the target classes "castle", "street sign", and "station wagon". While gradient-based methods can't distinguish between positive and negative contributions on the pixel level, explanations change significantly between classes and their highest values visibly lie within the object of the selected class. Heatmaps are computed on VGG-19 with $n = 10$ and $\sigma = 0.5$.

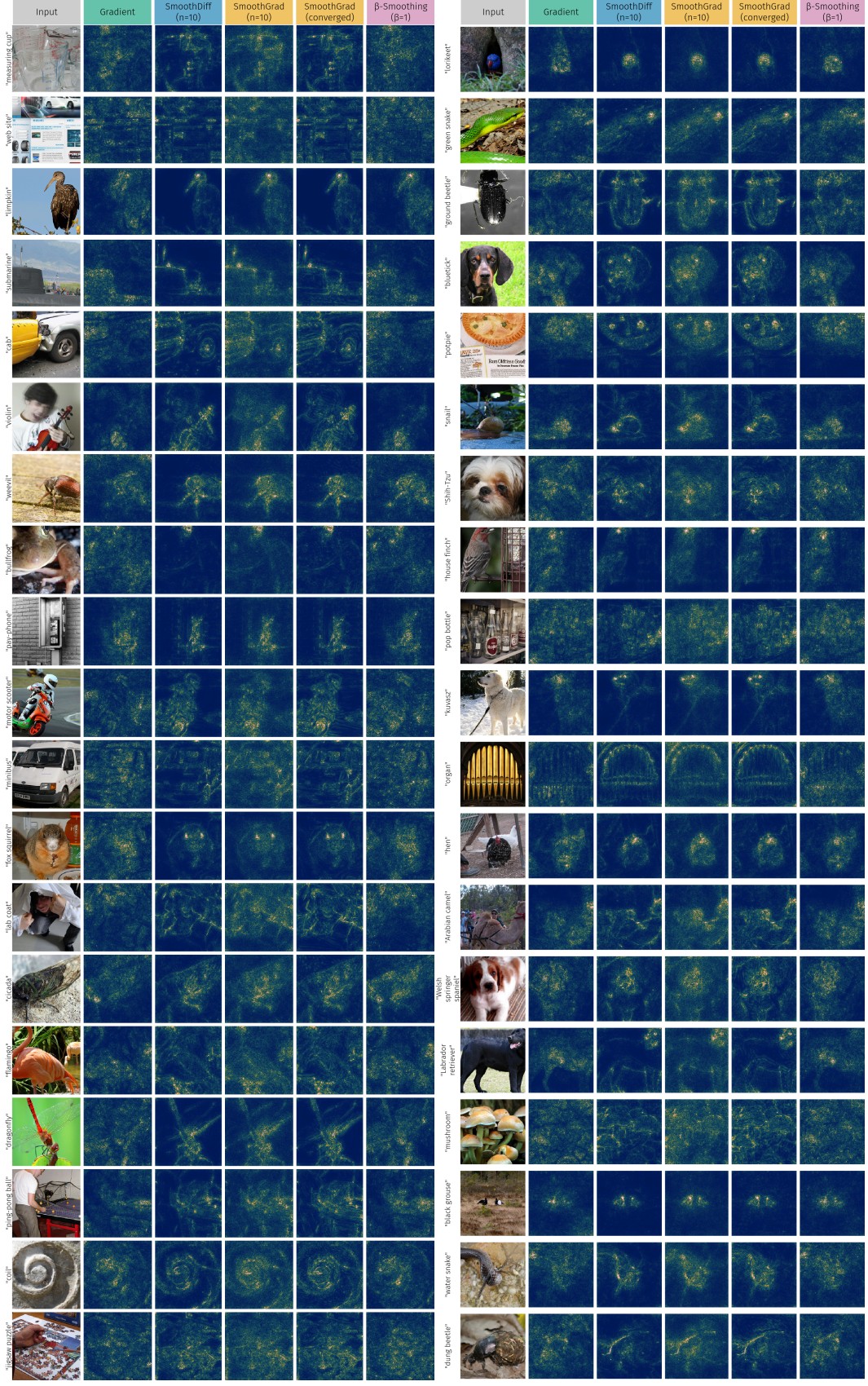

Figure 17: Comparison of gradient-based explanations on VGG-19. All SmoothDiff and Smooth-Grad explanations use $\sigma = 0.5$.

