# OpenReview forum: "Smoothed Differentiation Efficiently Mitigates Shattered Gradients in Explanations"
_NeurIPS.cc/2025/Conference — NeurIPS 2025 poster_

### Official Review · Reviewer_WSGh · 2025-06-30

**Clarity:** 3
**Significance:** 3
**Originality:** 3
**Rating:** 4
**Confidence:** 4

**Summary:**

This paper proposes a method to speed up Smooth Grad. The quality of smooth grad is depended on the number of input samples, which makes it slow in practice. Typically, at least 50 samples are required for the gradients to obtain reasonable quality. This paper proposes to  automatic differentiation to decompose the expected values of Jacobians across a network architecture, directly targeting only the non-linearities responsible for shattered gradients. The paper demonstrates speed-up over the default implementation of smooth grad.

**Questions:**

1) Figure 2 shows that you can reduce the number of samples from 50 to 10. This would give you a speed up of 5X. How do you get to two-three orders of speedup in the abstract?
2) Can the method be applied to a broad range of computer vision models?
3) Do the speed-ups hold for a broad range of attribution tests?

**Ethical Concerns:**

["NO or VERY MINOR ethics concerns only"]

**Final Justification:**

The authors have provided results that compare their methods with some SOTA methods. While SmoothGrad may be slightly outdated,  SmothDiff provides a non trivial speedup that could potentially be integrated into other methods. I agree with the other reviewers that this could be a valuable addition to the conference program. Therefore, I increased my score to a borderline accept.

**Limitations:**

Several limitations have been mentioned above.

**Quality:**

3

**Strengths And Weaknesses:**

Strength
1) The shattered gradient problem is well motivated.
2) Gaussian convolutions in the embedding space appears to reduce the impact of shattered gradients. It builds on the automated differentiation built into PyTorch.

Weaknesses:
1) Smoothgrad is hardly the SOTA method at this time. Therefore, the authors should show that Smoothgrad adds value compared with SOTA methods. For CNNs, comparisons could be provided with GradCAM and for ViTs, comparisons could be provided with rollout and subsequent work.
2) There is no evaluations on modern neural network architectures such as ViTs
3) There has been a number of different sanity tests and evaluation metrics for attribution methods. Fidel-flipping seems to be arbitrary selected and does not provide a comprehensive evaluation.
4) The methodology seems to have been written with an unnecessary amount of math. It is very heavy to read and not easy to follow.

---

> ### Author Rebuttal · Authors · 2025-07-31
>
> We thank the reviewer for their time and suggestions.
>
> Before we address the feedback, we want to highlight that due to a typo, Fig. 1b visualized SmoothGrad with the wrong sample size.
> We caught this error while working on the supplementary material and included an updated version in Appendix I.
>
> ## Weaknesses
> > Smoothgrad is hardly the SOTA method at this time. Therefore, the authors should show that Smoothgrad adds value compared with SOTA methods. For CNNs, comparisons could be provided with GradCAM
>
> > There has been a number of different sanity tests and evaluation metrics for attribution methods. Fidel-flipping seems to be arbitrary selected and does not provide a comprehensive evaluation.
>
> We expanded our quantitative evaluation to include **nine XAI methods** (SmoothDiff, SmoothGrad, Gradient, GradientShap, GradCAM, Integrated Gradients, Input x Gradient, two LRP composites) and **six evaluation metrics**.
> We selected the five metrics proposed by Bommer et al. (2024), as well as the *Pointing Game*:
>
> * **Robustness:** Avg-Sensitivity (Yeh at el., 2019), measures the change of attributions under slight input perturbations.
> * **Faithfulness:**  Faithfulness Correlation (Bhatt et al., 2020), flips of a random subset of pixels and computes the Pearson correlation coefficient between the predicted logits and average attribution for the flipped features.
> * **Localization:**  Relevance Rank Accuracy (Arras et al., 2021), measures the ratio of high attribution pixels within a provided ground truth mask.
> * **Complexity:**  Sparseness (Chalasani et al., 2020), computes the Gini index applied to the absolute values of attributions, indicating that an attribution consists of only a few important features.
> * **Randomization:**  Random Logit Metric (Sixt et al., 2020), computes the distance between an attribution and the attribution of a randomly selected non-target class.
> * **Pointing Game:**  (Zhang et al., 2018), check whether the point of maximal attribution lies within a provided ground truth mask.
>
> Reference implementations from Captum (Kokhlikyan et al., 2022) are used for all XAI methods except LRP, for which composites from Zennit (Anders et al., 2021) are used.
> Metrics are evaluated on a pre-trained ResNet-18 using the the Quantus toolkit (Hedström et al., 2024) with default parameters.
> For SmoothDiff, SmoothGrad, Integrated Gradients, and GradientShap, attributions are computed using 20 samples.
>
> Since this year's NeurIPS rebuttal policies prohibit the inclusion of PDFs, images and URLs, we provide our results in form of tables.
> To improve the readability of our results, we report the ranking of a method within each metric, followed by the mean score in parenthesis.
> With the exception of the *Robustness* metric, a higher score is better.
>
> **Table R1: Results on ResNet-18**
> |**Method**|**Robustness**|**Faithfulness**|**Localization**|**Complexity**|**Randomization**|**Pointing Game**|
> |:-|:-|:-|:-|:-|:-|:-|
> |SmoothDiff|**1 (0.45)**|3 (-0.016)|**1 (0.62)**|6 (0.430)|2 (0.51)|**1 (0.82)**|
> |SmoothGrad|2 (0.57)|2 (-0.012)|2 (0.61)|8 (0.397)|8 (0.41)|3 (0.65)|
> |Gradient|3 (0.74)|7 (-0.060)|4 (0.55)|7 (0.425)|5 (0.43)|2 (0.71)|
> |GradientShap|5 (1.19)|8 (-0.066)|7 (0.51)|5 (0.583)|6 (0.43)|3 (0.65)|
> |Integrated Gradients|4 (1.15)|6 (-0.059)|6 (0.51)|4 (0.584)|7 (0.42)|3 (0.65)|
> |Input x Gradient|6 (1.24)|5 (-0.039)|8 (0.51)|3 (0.597)|3 (0.45)|3 (0.65)|
> |LRP EpsilonPlus|7 (1.46)|4 (-0.019)|3 (0.58)|2 (0.608)|4 (0.43)|8 (0.53)|
> |LRP EpsilonAlpha2Beta1|8 (1.64)|**1 (-0.005)**|5 (0.54)|**1 (0.754)**|**1 (0.82)**|3 (0.65)|
>
> On ResNet-18, SmoothDiff has the highest *Robustness* w.r.t. input perturbations, while the high *Faithfulness* score mirrors the favorable pixel-flipping performance reported in Section 4.4.
> The *Randomization* score demonstrates that SmoothDiff attributions strongly depend on the selected output class.
> SmoothDiff performs well in the *Pointing Game* and the *Localization* metric, both of which measure the localization of attributions within ground truth masks.
> **With the exception of *Sparseness*, SmoothDiff places in the top 3 on all metrics, while being easy to implement**.
> Since attributions *qualitatively* look sparse, we will further investigate this metric for the camera-ready version.
>
> On VGG19 (see Table R2), SmoothDiff performs similarly well, showing that these results are not model dependant.
>
> **Table R2: Results on VGG19**
> |**Method**|**Robustness**|**Faithfulness**|**Localization**|**Complexity**|**Randomization**|**Pointing Game**|
> |:-|:-|:-|:-|:-|:-|:-|
> |SmoothDiff|**1 (0.45)**|2 (0.012)|4 (0.58)|7 (0.50)|3 (0.60)|**1 (0.65)**|
> |SmoothGrad|2 (0.57)|3 (0.006)|3 (0.59)|9 (0.44)|9 (0.52)|6 (0.59)|
> |Gradient|3 (0.79)|4 (0.001)|2 (0.60)|8 (0.48)|6 (0.55)|**1 (0.65)**|
> |GradCAM|4 (1.00)|5 (0.001)|9 (0.51)|**1 (0.85)**|2 (0.76)|**1 (0.65)**|
> |GradientShap|7 (1.25)|8 (-0.049)|7 (0.52)|6 (0.62)|8 (0.54)|6 (0.59)|
> |Integrated Gradients|5 (1.21)|7 (-0.022)|6 (0.52)|5 (0.62)|7 (0.55)|9 (0.53)|
> |Input x Gradient|6 (1.23)|9 (-0.056)|8 (0.52)|4 (0.64)|4 (0.58)|**1 (0.65)**|
> |LRP EpsilonPlus|8 (1.52)|**1 (0.022)**|**1 (0.66)**|3 (0.64)|5 (0.58)|**1 (0.65)**|
> |LRP EpsilonAlpha2Beta1|9 (5.76)|6 (-0.007)|5 (0.57)|2 (0.80)|**1 (0.86)**|6 (0.59)|
>
> > There is no evaluations on modern neural network architectures such as ViTs
>
> As we explain in Section 3.1, the performance of SmoothDiff highly depends on the selected Jacobian factorization.
> While SmoothDiff can be applied to arbitrary models, including ViTs, we have not yet found a factorization for which SmoothDiff outperforms SmoothGrad.
> We think that SmoothDiff's quantitative results on CNN (see Tables R1, R2) stand on their own, especially in light of how easy the method is to implement.
> We will update the "Limitations" section of our camera-ready version to explicitly mention the current lack of performant vector-Jacobian products (VEJPs) for ViTs.
>
> Something we have yet to investigate is whether a partial application of SmoothDiff improves attributions over "vanilla" gradients. To provide an example in the context of ViTs, VEJPs could only be implemented on GeLU, computing "vanilla" gradients on all other non-linear layers.
>
> > The methodology seems to have been written with an unnecessary amount of math. It is very heavy to read and not easy to follow.
>
> The first five equations are essential to the paper:
> Eq. 1-3 introduce the preliminaries SmoothDiff builds upon (Gaussian convolutions, SmoothGrad, Jacobian factorizations and vector-Jacobian products) and in Eq. 4-5, we propose SmoothDiff itself.
>
> The remaining four equations are meant to serve as illustrative practical examples: Eq. 6 shows the Jacobian factorization on a Dense layer and Eq. 7-9 show the derivation of a VEJP on a ReLU.
>
> ## Questions
>
> > Figure 2 shows that you can reduce the number of samples from 50 to 10. This would give you a speed up of 5X.
>
> Figure 2 shows SmoothDiff's 5X increase in sample efficiency.
> In addition, SmoothDiff has a purely computational speedup of 2X at *equal* sample sizes, as shown in Section 4.3, Figure 4.
>
> > How do you get to two-three orders of speedup in the abstract?
>
> The computational speedup of two orders is described in **Section 3.3 "Computational performance", lines 187-194**:
> Once VEJPs have been estimated in $n$ forward-passes, SmoothDiff attributions can be computed in a single backward-pass. Computing multi-class attributions for $k$ classes (or hidden neurons / concepts) only takes $k$ backward-passes, for a total of $n+k$ forward- and backward passes.
> SmoothGrad on the other hand requires $n$ forward- and $n$ backward-passes for each attribution, resulting in a total of $2nk$ passes through the model.
>
> Assuming an attribution of all $k=1000$ ImageNet classes ($k$ could be even higher for concept attributions) using $n=50$ samples, SmoothGrad requires $2nk=10^5$ passes through the model, whereas SmoothDiff only requires $n+k=1050$.
> This corresponds to a purely computational speedup of two orders of magnitude, ignoring the increased sample efficiency.
>
> > Can the method be applied to a broad range of computer vision models?
>
> Yes, SmoothDiff can be applied to arbitrary differentiable models, including computer vision models.
> For any function (e.g. a layer), a "naive" VEJP can be implemented that uses automatic differentiation (AD) to compute the full Jacobian on each forward-pass, keeping track of a running mean.
> However, for large and dense Jacobians, this "naive" approach requires a lot of memory and compute, negating SmoothDiff's performance benefits over SmoothGrad.
>
> As we discuss in Section 3.1, the performance of SmoothDiff therefore strongly depends on the factorization.
> The reported computational speed-ups of 2X hold for both VGG and ResNet models.
> For the camera-ready version, we will update Subsection 3.2 to include the naive running mean VEJP and the "Limitations" sections to explicitly mention the current lack of performant factorizations for ViT.
>
> > Do the speed-ups hold for a broad range of attribution tests?
>
> SmoothDiff's speed-ups (both computational and in sample efficiency) hold independent of evaluation metrics for attributions.

---

> > ### Comment · Reviewer_WSGh · 2025-08-05
> > **Additional experimental results**
> >
> > I appreciate the updated experimental results section. The paper clearly shows that the proposed framework is an improvement over SmoothGrad. However, SmoothGrad is not a SOTA method for attributions anymore, which is evidenced by the results that have been provided in the rebuttal. Also, It would have been good to include perturbation games such as Insertion and Deletion in the evaluation, which are very common technique for evaluating attribution methods. Nevertheless, I will raise my score to a weak reject as the positioning of the paper with SOTA methods is now more clear.

---

> ### Comment · Reviewer_Meht · 2025-08-05
> **Question about SOTAs**
>
> Thanks for your review and the raised points, which I found quite insightful.
>
> Could you kindly elaborate on what are the SOTA methods that would be good to compare against? As far as I know, LRP or Integrated Gradients that the authors already compare to in Tables R1&R2, are among the stronger attribution methods. And it seems that SmoothDiff raises SmoothGrad to be ranked 1st among some of the metrics (e.g. Pointing Game). I have asked the authors to maybe compare with inherently-interpretable models (e.g. x-DNNs or B-cos) as a sort of "upper bound" and hard baselines. But I'm curious if you find any crucial attribution method missing from Tables R1&R2.
>
> Thanks!

---

> ### Author Response · Authors · 2025-08-05
> **Question regarding missing "SOTA method"**
>
> We thank the reviewer for engaging with our rebuttal.
>
> > I appreciate the updated experimental results section. The paper clearly shows that the proposed framework is an improvement over SmoothGrad.
>
> Our results show that on ResNets, SmoothDiff **not only outperforms SmoothGrad, but places in the top 3 in five of the six evaluated metrics**, while being easy to implement.
>
> In their initial feedback, the reviewer asked for a comparison with GradCAM as the SOTA method on CNN:
>
> >> Smoothgrad is hardly the SOTA method at this time. Therefore, the authors should show that Smoothgrad adds value compared with SOTA methods. For CNNs, comparisons could be provided with GradCAM
>
> We added this requested comparison with GradCAM in our initial rebuttal.
> Our results on VGG19 show that the proposed SmoothDiff method outperforms GradCAM in terms of *Robustness*, *Faithfulness*, *Localization* and draws first place in the *Pointing Game*.
>
> > However, SmoothGrad is not a SOTA method for attributions anymore, which is evidenced by the results that have been provided in the rebuttal.
>
> We reemphasize that SmoothDiff not only outperforms SmoothGrad, but also state-of-the art LRP composites and GradCAM (the proposed SOTA method on CNN) on several metrics.
>
> We agree with the sentiment expressed by reviewer "Meht" and **kindly ask reviewer "WSGh" to elaborate on the additional SOTA methods they would like to see evaluated**.
>
> > Also, It would have been good to include perturbation games such as Insertion and Deletion in the evaluation, which are very common technique for evaluating attribution methods.
>
> We provide several evaluation metrics based on perturbations:
> * Section 4.4 of our paper includes detailed pixel-flipping results based on the novel *symmetric relevance gain* measure introduced by Bluecher et al. (2024), which removes the influence of infilling and in-painting methods (i.e. insertions and deletions) on pixel-flipping.
> * The *Robustness* metric *Avg-Sensitivity* (Yeh at el., 2019) measures the change of attributions under slight input perturbations.
> * *Faithfulness Correlation* (Bhatt et al., 2020) flips a random subset of pixels and computes the Pearson correlation coefficient between the predicted logits and the average attribution for the flipped features.
>
> We are willing to extend our evaluation to further perturbation-based metrics if the reviewer elaborates on which metrics they think are missing.

---

> > ### Comment · Reviewer_WSGh · 2025-08-06
> > **SOTA Methods**
> >
> > My concern regrading the lack of comparisons with SOTA methods was prompted by that all the cited methods (IG, GradCam, LRP, etc) were published in the 2016 to 2019 timeframe. A substantial number of papers have been published on this topic in the 2020-2025 timeframe. A few samples are listed below. It would be more convincing if comparisons were provided with those or any other methods from the past two years.
> >
> > In terms of metrics, the authors have omitted the standard insertion and deletion tests where pixels are gradually inserted or deleted. These are the tests typically used when assessing the quality of attribution methods. While most metrics for assessing attribution methods have some limitations, those tests provide some valuable insights regarding the quality of the attribution method. Moreover, many works are geared towards archiving good scores on those tests.
> >
> > [1] Adversarial Integrated Gradients: https://www.ijcai.org/proceedings/2021/0396.pdf
> > [2] Yang, P.; Akhtar, N.; Wen, Z.; and Mian, A. 2023. Local path
> > integration for attribution. In Proceedings of the AAAI Con-
> > ference on Artificial Intelligence, volume 37, 3173–3180.
> > [3] Yang, R.; Wang, B.; and Bilgic, M. 2023. IDGI: A Frame-
> > work to Eliminate Explanation Noise from Integrated Gradi-
> > ents. In Proceedings of the IEEE/CVF Conference on Com-
> > puter Vision and Pattern Recognition, 23725–23734.

---

> > > ### Author Response · Authors · 2025-08-06
> > > **Deletion and Insertion tests are already included**
> > >
> > > > My concern regrading the lack of comparisons with SOTA methods was prompted by that all the cited methods (IG, **GradCam**, LRP, etc) were published in the 2016 to 2019 timeframe.
> > >
> > > We want to politely remind the reviewer that they explicitly asked for evaluations on GradCAM, which they considered the SOTA method on CNN at the time of their review.
> > >
> > > Since we added GradCAM evaluations to our rebuttal based on their request, we are surprised by their change of mind regarding the significance of these results.
> > >
> > > > A substantial number of papers have been published on this topic in the 2020-2025 timeframe. A few samples are listed below. It would be more convincing if comparisons were provided with those or any other methods from the past two years.
> > >
> > > We thank the reviewer for providing pointers to additional methods they consider SOTA.
> > > We will try our best to evaluate them within the limited amount of time left in the "Reviewer Author Discussions" phase.
> > >
> > > > In terms of metrics, the authors have omitted the standard insertion and deletion tests where pixels are gradually inserted or deleted. These are the tests typically used when assessing the quality of attribution methods.
> > >
> > > **These tests have not been omitted.**
> > > As we have explained to the reviewer in our comment titled *Question regarding missing "SOTA method"*, section 4.4 of our paper already includes these insertion and deletion tests. The *Symmetric Relevance Gain* measure (Bluecher et al., 2024) computes two curves:
> > >
> > > * By deleting pixels in order of *most-influential features* and computing the area under the resulting prediction curve, we obtain the *MIF measure*, which is **equivalent to the deletion score** in the provided reference [1].
> > > * By deleting pixels in order of *least-influential features* and computing the area under the resulting prediction curve, we obtain the *LIF measure*, which is **equivalent to the insertion score** in the provided reference [1].
> > > * The *Symmetric Relevance Gain* additionally computes the difference between the LIF and MIF measures, corresponding to the area between the insertion and deletion curves.
> > >
> > > We report all three scores (LIF / insertion, MIF / deletion, SRG) in Section 4.4, Table 1.
> > > We will update the table with methods from the rebuttal for the camera-ready version of our paper.

---

### Official Review · Reviewer_Meht · 2025-07-01

**Clarity:** 3
**Significance:** 3
**Originality:** 3
**Rating:** 4
**Confidence:** 3

**Summary:**

The paper proposes a more efficient way to approximate the Jacobian of a network over different samples (sampling done for having less noise in the final attribution). The empirical results show that the proposed method has more efficient sampling, and for the same # samples or wall-time, is able to converge faster.

The method is explored for CNNs (ResNet and VGG) and on ImageNet. The final attributions are evaluated in terms of SSIM/#samples (for convergence comparison) Wall-time/#samples (for efficiency) and Pixel-flip/#samples for faithfulness and overall quality of explanations. There is also qualitative comparison to SmoothGrad (prior work with less efficient sampling) and also vanilla grad (no sampling → noisy).

**Questions:**

Could the authors clarify if the final attribution map would be differentiable, and whether this also is the case for the compared baseline (SmoothGrad)? This is not a major point for me, but it is good to know, as there are works that use such attribution maps to guide the models.

See also the weaknesses.

**Ethical Concerns:**

["NO or VERY MINOR ethics concerns only"]

**Final Justification:**

The rebuttal thoroughly addressed my concerns, and the promised version of the manuscript is in my opinion a good paper. I agree with Reviewer WSGh that Smooth Grad may not be THE most exciting attribution method that one wants to improve upon, but this paper shows that by improving it in terms of sample efficiency, it does indeed become quite competitive to other widely used methods such as Integrated Gradients and LRP. I believe the promised revision would be a good contribution to the community.

**Limitations:**

There are more limitations in terms of applicability to currently used models (see concern (d) above). These should also be discussed in the limitations.

**Paper Formatting Concerns:**

Yes, except that the bib entries are not numbered.

**Quality:**

3

**Strengths And Weaknesses:**

**[Strengths]**

I think overall this is a solid paper with clear discussion followed by empirical results to support the claims. I was able to mostly follow the derivation and figures. To me the idea of not estimating the Jacobian in one-go and rather engineering it based on the components with different rules makes sense.

The quantitative results seem convincing, although there are many things missing (see weaknesses).

The qualitative results (e.g. Fig 3) was also convincing to me.

**[Weaknesses]**

**(a) Writing**

I would say the paper is mostly well-written, as the paragraphs are easy to follow. However, the overall structure is poor. For example, in the introduction, instead of motivating the problem setup (sampling), the importance of attributions, their use-cases and the importance of their quality, or why one would need attributions of different classes, the paper instead goes through the related work in very high detail. Basically from Lines 31 to 82, all of this should be spread to the Related Work section or preliminaries in the Method section.

The captions on the figures are also very uninformative. The caption should guide the reader on the trend and main message of the figure, not to only repeat what is already written on the axis or legend.

I also think there is a formatting issue with the citations not having numbers in the bib.

**(b) Quantitative Evaluation of Attributions**

I appreciate the author including the pixel flipping study. But by now there are many more ways of evaluating the attributions, and I don’t see why the paper should confine itself to only one. For example, [1] proposes Grid-Pointing Game, specifically targeting the localization and interpretability of the explanations. [2] proposes a new way of evaluating attribution methods. [3] proposes a synthetic benchmark for faithfulness of such methods, or [4] proposes sanity checks for faithfulness. Especially given the qualitative results, I expect Grid-Pointing Game score to be much higher than baselines, and this would further ensure that the qualitative observation (of less-noise for same # samples) indeed is supported quantitatively.

**(c) Baseline Attribution Methods**

I understand why the authors mainly compare to SmoothGrad, as this is one of the closest baselines, but in the end, the community and people outside XAI community would simply need an attribution map for the decision. Therefore, it is necessary to put the proposed attribution method in comparison with more attributions. E.g. LRP, Integrated Gradients. Ideally, one should also go with other methods providing efficient explanations, such as [5], where for the proposed x-DNN architecture the InputxGrad is equivalent to integrated gradients.

I also was not able to follow the discussion on Lines 186-186. Is SmoothDiff orthogonal to LRP? How would LRP benefit from SmoothDiff exactly? As far as I know LRP has its own custom rules of recursively assigning relevance, so I don’t see how efficient sampling would help LRP?

Without any quantitative analysis on such well-established related work, (especially with metrics such as the ones I said above), it is difficult for me to accept the paper.

**(d) Scope**

There are many commonly used architectures that are significantly different from this paper. We of course have ViTs, with multi-head self-attention and softmax, ConvNext with GeLU instead of ReLU and many more. To me it is acceptable if the proposed approach does not readily extend to this, but the lack of discussion on this matter concerns me. Especially because one can think of other attribution methods such as IntGrad, Vanilla Grad, and even SmoothGrad (maybe?), to be superior in that case, by abstracting away from individual layer/operation types and readily extending to such architectures. I think the authors should clearly discuss how well their method extends to other architectures, both for the revisions and in the rebuttal.

[1] CoDA Nets, CVPR 2021, https://arxiv.org/abs/2104.00032

[2] Evaluating Attributions, CVPR 2022, https://arxiv.org/abs/2205.10435

[3] Funny Birds, ICCV 2023, https://arxiv.org/abs/2308.06248

[4] Sanity Checks, NeurIPS 2018, https://arxiv.org/abs/1810.03292

[5] x-DNNs, NeurIPS 2021, https://arxiv.org/abs/2111.07668

---

> ### Author Rebuttal · Authors · 2025-07-30
>
> We thank the reviewer for their time and expertise, as well as for underlining the quality of our approach.
>
> Before addressing the feedback, we want to highlight that due to a typo, Fig. 1b visualized SmoothGrad with the wrong sample size. We caught this error while working on the supplementary material and included an updated version in Appendix I.
>
> ## Weaknesses
> ### (a) Writing
> > I would say the paper is mostly well-written, as the paragraphs are easy to follow. However, the overall structure is poor [...]
> > Basically from Lines 31 to 82, all of this should be spread to the Related Work section or preliminaries in the Method section.
>
> Since this year's NeurIPS rebuttal policies prohibit the inclusion of PDFs, URLs and images, we will address these issues in the camera-ready version of the paper.
> We will restructure the "Introduction" and "Related Work" sections, provide more detail on both the problem setup and desired properties (see (b) below) and add a subsection for preliminaries.
>
> > The captions on the figures are also very uninformative.
>
> We will provide more detailed captions in the camera-ready version of the paper.
>
> > I also think there is a formatting issue with the citations not having numbers
>
> We have fixed this issue.
>
> ### (b) Quantitative Evaluation of Attributions
> > I appreciate the author including the pixel flipping study. But by now there are many more ways of evaluating the attributions, and I don’t see why the paper should confine itself to only one. For example, [1] proposes Grid-Pointing Game, specifically targeting the localization and interpretability of the explanations.
>
> We appreciate the suggestion and pointers to relevant metrics. Within the time frame of the rebuttal, we have opted to use the Quantus toolkit (Hedström et al., 2024), which provides a common interface for 35+ metrics, including the proposed pointing game. We additionally evaluated the five metrics proposed by Bommer et al. (2024):
>
> * **Robustness:** Avg-Sensitivity (Yeh at el., 2019), measures the change of attributions under slight input perturbations.
> * **Faithfulness:**  Faithfulness Correlation (Bhatt et al., 2020), flips of a random subset of pixels and computes the Pearson correlation coefficient between the predicted logits and average attribution for the flipped features.
> * **Localization:**  Relevance Rank Accuracy (Arras et al., 2021), measures the ratio of high attribution pixels within a provided ground truth mask.
> * **Complexity:**  Sparseness (Chalasani et al., 2020), computes the Gini index applied to the absolute values of attributions, indicating that an attribution consists of only a few important features.
> * **Randomization:**  Random Logit Metric (Sixt et al., 2020), computes the distance between an attribution and the attribution of a randomly selected non-target class.
>
> Captum (Kokhlikyan et al., 2022) is used for all XAI methods except LRP, for which Zennit (Anders et al., 2021) is used. Metrics are evaluated on a pre-trained ResNet-18 using default parameters. For SmoothDiff, SmoothGrad, Integrated Gradients, and GradientShap, attributions are computed using 20 samples. To improve the readability of our results, we report the ranking of a method within each metric, followed by the mean score in parenthesis. With the exception of the *Robustness* metric, a higher score is better.
>
> |**Method**|**Robustness**|**Faithfulness**|**Localization**|**Complexity**|**Randomization**|**Pointing Game**|
> |:-|:-|:-|:-|:-|:-|:-|
> |SmoothDiff|**1 (0.45)**|3 (-0.016)|**1 (0.62)**|6 (0.430)|2 (0.51)|**1 (0.82)**|
> |SmoothGrad|2 (0.57)|2 (-0.012)|2 (0.61)|8 (0.397)|8 (0.41)|3 (0.65)|
> |Gradient|3 (0.74)|7 (-0.060)|4 (0.55)|7 (0.425)|5 (0.43)|2 (0.71)|
> |GradientShap|5 (1.19)|8 (-0.066)|7 (0.51)|5 (0.583)|6 (0.43)|3 (0.65)|
> |Integrated Gradients|4 (1.15)|6 (-0.059)|6 (0.51)|4 (0.584)|7 (0.42)|3 (0.65)|
> |Input x Gradient|6 (1.24)|5 (-0.039)|8 (0.51)|3 (0.597)|3 (0.45)|3 (0.65)|
> |LRP EpsilonPlus|7 (1.46)|4 (-0.019)|3 (0.58)|2 (0.608)|4 (0.43)|8 (0.53)|
> |LRP EpsilonAlpha2Beta1|8 (1.64)|**1 (-0.005)**|5 (0.54)|**1 (0.754)**|**1 (0.82)**|3 (0.65)|
>
> SmoothDiff (SD) has the highest *Robustness* w.r.t. input perturbations, while the high *Faithfulness* score mirrors the favorable pixel-flipping performance reported in Section 4.4. The *Randomization* score demonstrates that SD attributions strongly depend on the selected output class. With the exception of *Sparseness*, SD places in the top 3 on all metrics, **while being easy to implement**. Since attributions *qualitatively* look sparse, we will further investigate this for the camera-ready version.
>
> > I expect Grid-Pointing Game score to be much higher than baselines, and this would further ensure that the qualitative observation (of less-noise for same # samples) indeed is supported quantitatively.
>
> The results indeed show that SD performs well in the *Pointing Game* and the *Localization* metric, both of which measure the localization of attributions within ground truth masks.
>
> > [2] proposes a new way of evaluating attribution methods. [3] proposes a synthetic benchmark for faithfulness of such methods, or [4] proposes sanity checks for faithfulness.
>
> While those benchmarks fell outside of the time scope of this rebuttal, we will include references to these metrics in the camera-ready.
>
> ### (c) Baseline Attribution Methods
> > it is necessary to put the proposed attribution method in comparison with more attributions. E.g. LRP, Integrated Gradients.
>
> We expanded our quantitative evaluation to include 9 methods (SD, SmoothGrad, Gradient, GradientShap, GradCAM, Integrated Gradients, IxG, two LRP composites), 2 models (VGG19, ResNet-18) and 6 metrics. The results on ResNet-18 are shown in the table above and results on VGG19 can be provided during the reviewer-author discussion (due to the character limit).
>
> > I also was not able to follow the discussion on Lines 186-186. Is SmoothDiff orthogonal to LRP? How would LRP benefit from SmoothDiff exactly?
>
> LRP is computationally very efficient, computing attributions in a single backward-pass through the model.
> We argue that the methods enumerated in L183-185 are enabled by LRP's computational efficiency and would not be feasible using SmoothGrad. As an example, computing attributions w.r.t. $k$ hidden neurons in *Concept Relevance Propagation* (Achtibat et al., 2023) only requires $k$ backward-passes, whereas SmoothGrad with $n$ samples requires $nk$ passes.
>
> SD is also computationally very efficient: Once the vector-expected-Jacobian product (VEJP) operator has been estimated in $n$ forward-passes, attributions can be computed in a single backward-pass. Computing attributions for $k$ classes / hidden neurons / concepts only takes $k$ backward-passes, like LRP.
>
> We will clarify this in the camera-ready, as it is a key strength of SmoothDiff.
>
> ### (d) Scope
> > There are many commonly used architectures that are significantly different from this paper. We of course have ViTs, with multi-head self-attention and softmax, ConvNext with GeLU instead of ReLU and many more.
>
> SD can be applied to arbitrary models and factorizations, but the performance will vary.
>
> For any function (e.g. a layer), a "naive" VEJP can be implemented that uses automatic differentiation (AD) to compute the full Jacobian on each forward-pass, keeping track of a running mean.
> For a GeLU $f(x)=x \Phi(x)$, this approach is very performant, only requiring $n$ evaluations of $f'(x)=x \Phi'(x) + \Phi(x)$, one for each sample.
> However, for very large and dense Jacobians (e.g. attention blocks in ViT), this "naive" approach requires a lot of memory and compute, negating SD's performance benefits over SmoothGrad.
>
> Something we have yet to investigate is whether a partial application of SD improves attributions over "vanilla" Gradients. To provide an example in the context of ViT, VEJP could only be implemented on GeLU, computing "vanilla" gradients on all other non-linear layers.
>
> > To me it is acceptable if the proposed approach does not readily extend to this, but the lack of discussion on this matter concerns me.
>
> We fully agree: for the camera-ready version, we will update the section on "Factorization" to include the naive running mean VEJP and the "Limitations" sections to explicitly mention the lack of performant VEJP for ViT.
>
> ## Questions
> > Could the authors clarify if the final attribution map would be differentiable, and whether this also is the case for the compared baseline (SmoothGrad)?
>
> This is a great question. Differentiating the gradient of a model would result in its Hessian, which describes the curvature of the model at a given input. Deep ReLU networks are piece-wise linear and have no curvature. Their gradients are piecewise constant and their Hessians therefore zero. However, by applying a Gaussian convolution to the gradient, curvature is introduced (see Fig. 1a). Mathematically speaking, the function obtained by applying a convolutional operator to the gradient function is therefore differentiable (with non-zero derivatives).
>
> Implementation-wise, Hessians are usually computed via AD by nesting forward- and reverse-mode to evaluate a series of Hessian-vector products (see Dagréou et al., 2024). As we discuss in the paper, SD can be seen as a modified reverse-mode AD pass. It could therefore potentially be used within the computation of an "smoothed" HVP. While we don't feel confident enough to make claims about the practical differentiability of SD (outside of numerical approximations like finite differences), we will update the outlook of the camera-ready to mention 2nd-order explanations.
>
> ## Limitations
> > There are more limitations in terms of applicability to currently used models (see concern (d) above). These should also be discussed in the limitations.
>
> We have addressed this in our rebuttal of concern (d) and will update the limitations section of our camera-ready version.

---

> ### Comment · Reviewer_Meht · 2025-08-04
> **Mostly Convinced with the Rebuttal**
>
> I would like to thank the authors for their detailed and clear response. I'm mostly convinced that the promised revision of the manuscript will be stronger. There are however three more questions I would like to ask.
>
> **a)** Could the authors however elaborate on the Localization Metric? It is slightly strange to me that the Integrated Gradients is worse than vanilla gradient. Could you elaborate on how many samples was used for Integrated Gradients? Also is there any random baseline number for this metric?
>
> **b)** Additionally, I am generally against comparing attribution methods with human-annotations, as it is not guaranteed that the model should be using what we as humans expect. If the authors could provide an evaluation using GridPG [1,2], which creates a 2x2 grid of images with different labels and measures how much of the attribution of a class lies in the correct cell, which is therefore annotation-free, that would be even more convincing to me. I checked and there are multiple implementations of this available [3,4]. I understand that the time is limited, but this would strengthen the evaluation in my opinion.
>
> If their results are compared to model-inherent methods such as x-DNN or B-cos, which are more recent, then this would also partially address Weakness 1 raised by Reviewer WSGh. I don't necessarily expect this method to beat model-inherent attributions, but a comparison to such methods would put this method on the (more recent) grid.
>
> **c)** Lastly, have the authors done any experiments on a multi-label dataset, or simply images that have multiple classes, to see whether the multi-class attributions are valid? For example, an image with Person + Bike from Pascal VOC dataset, where the attribution for Person and Bike are estimated? I think this would help the paper, even if done only on a qualitative level, given that the goal is to estimate the Jacobian more efficiently. One can also do this with an ImageNet model, just using images that have more than one category in them.
>
>
> [1] CoDA Nets, CVPR 2021, https://arxiv.org/abs/2104.00032
>
> [2] B-cos Alignment for Inherently Interpretable CNNs and Vision Transformers, https://arxiv.org/abs/2306.10898
>
> [3] https://github.com/B-cos/B-cos-v2/blob/main/interpretability/analyses/localisation.py
>
> [4] https://github.com/sukrutrao/Attribution-Evaluation/blob/master/scripts/evaluate_localization.py

---

> > ### Author Response · Authors · 2025-08-05
> > **Regarding the localization of Integrated Gradients**
> >
> > We thank the reviewer for their positive response to our rebuttal and for taking the time to engage with it.
> >
> > The purpose of this response is to address points (a) and (c) in a timely manner, allowing further discussion.
> > We also like to thank the reviewer for their pointers to open source implementations of *GridPG*, which we are currently working on evaluating. We will provide an update with our findings regarding point (b) in a separate response.
> >
> > > a) Could the authors however elaborate on the Localization Metric? It is slightly strange to me that the Integrated Gradients is worse than vanilla gradient. Could you elaborate on how many samples was used for Integrated Gradients?
> >
> > We used Captum's `IntegratedGradients` with 20 samples, a baseline of zero (`baselines=torch.zeros_like(inputs)`) and `method="riemann_trapezoid"` for integration.
> > We have opted to use the same number of samples for all sample-based methods in an attempt to make them comparable.
> >
> > The localization metric is the *Relevance Rank Accuracy* (RRA) by Arras et al. (2021).
> > We use Quantus' RRA implementation with default parameters on a set of 17 inputs and ground truths masks provided by Quantus. We will expand the sample count for the camera-ready version. Arras et al. describe RRA as follows:
> >
> > >> [RRA] measures how much of the high intensity relevances lie within the ground truth. It is calculated through the following steps. Let $K$ be the size of the ground truth mask. Get the $K$ highest relevance values. Then count how many of these values lie within the ground truth pixel locations, and divide by the size of the ground truth.
> >
> > Most XAI methods have in common that the dimensionality of their attributions matches the input. Hence, for ImageNet inputs, attributions contain three color channels (R, G, B).
> > However, metrics such as the RRA, which use single-channel ground truth masks, only measure the spatial location of pixel-wise attributions rather than that of individual color channels.
> > The RRA therefore requires a "reduction" of the three R, G, B attributions into a single scalar value (one per pixel). Arras et al. call this *relevance pooling* (RP).
> > As we describe in Appendix D, RP is also necessary for heatmapping, since only scalar values can be mapped onto a color scheme.
> >
> > As we motivate in our introduction, attributions can be based on different mathematical objects, e.g. sensitivities (e.g. Gradient, SmoothGrad, SmoothDiff) and Taylor expansions (e.g. IxG, LRP).
> > Within these families of methods, different conventions exist for RP:
> > * Methods based on Taylor expansions use summation, obtaining signed attributions. The distinction between positive and negative attributions is a key property of methods like LRP, resulting in their characteristic red-white-blue heatmaps.
> > * Methods based on sensitivities use the pixel-wise norm
> >
> > Since in terms of units, IG matches IxG and LRP, we initially ran our evaluation using summation for RP.
> > We have updated our evaluation to include IG with norm-RP:
> >
> > |**Method**|**Relevance Pooling**|**Robustness**|**Faithfulness**|**Localization**|**Complexity**|**Randomization**|**PointingGame**|
> > |:-|:-|:-|:-|:-|:-|:-|:-|
> > |SmoothDiff|norm|**1 (0.45)**|5 (-0.016)|**1 (0.615)**|7 (0.43)|3 (0.508)|**1 (0.824)**|
> > |SmoothGrad|norm|3 (0.57)|4 (-0.012)|2 (0.608)|9 (0.397)|9 (0.407)|3 (0.647)|
> > |Gradient|norm|5 (0.737)|9 (-0.06)|5 (0.553)|8 (0.425)|6 (0.432)|2 (0.706)|
> > |GradientShap|norm|7 (1.192)|10 (-0.066)|8 (0.509)|5 (0.583)|7 (0.427)|3 (0.647)|
> > |**Integrated Gradients**|**norm**|4 (0.733)|**1 (0.013)**|4 (0.557)|6 (0.502)|2 (0.55)|3 (0.647)|
> > |**Integrated Gradients**|**sum**|6 (1.15)|8 (-0.059)|7 (0.51)|4 (0.584)|8 (0.422)|3 (0.647)|
> > |Input x Gradient|sum|8 (1.24)|7 (-0.039)|9 (0.508)|3 (0.597)|4 (0.453)|3 (0.647)|
> > |LRP EpsilonPlus|sum|9 (1.459)|6 (-0.019)|3 (0.576)|2 (0.608)|5 (0.433)|10 (0.529)|
> > |LRP EpsilonAlpha2Beta1|sum|10 (1.642)|3 (-0.005)|6 (0.536)|**1 (0.754)**|**1 (0.819)**|3 (0.647)|
> > |**Random**|norm|2 (0.55)|2 (0.004)|10 (0.492)|10 (0.238)|10 (0.032)|9 (0.588)|
> >
> > As can be seen in the updated table, using norm-RP improves IG in terms of *Robustness*, *Faithfulness* and *Localization* (outperforming the vanilla gradient), but worsens the performance in terms of *Complexity* and *Randomization*.
> > We are open to suggestions on which type of RP should be used.
> >
> > > Also is there any random baseline number for this metric?
> >
> > We agree with the reviewer that the evaluation of a random baseline is valuable and have added it to the table above.
> > As expected, it performs badly in terms of *Localization*, *Complexity*, *Randomization* and the *Pointing Game*.
> >
> > Since the *Robustness* metric measures the change of attributions under slight input perturbations, random explanations perform well under this metric due to their input independence.
> > However, the results cast some doubt on the *Faithfulness Correlation* metric, which we now consider removing for the camera-ready version.

---

> > ### Author Response · Authors · 2025-08-05
> > **Regarding multi-class attributions**
> >
> > > c) Lastly, have the authors done any experiments on a multi-label dataset, or simply images that have multiple classes, to see whether the multi-class attributions are valid? For example, an image with Person + Bike from Pascal VOC dataset, where the attribution for Person and Bike are estimated? I think this would help the paper, even if done only on a qualitative level, given that the goal is to estimate the Jacobian more efficiently. One can also do this with an ImageNet model, just using images that have more than one category in them.
> >
> > As you allude to, multi-class attributions are a core strength of the proposed SmoothDiff method.
> > We describe performance improvements on multi-class attributions in depth in Section 3.3 "Computational performance".
> > Qualitatively speaking, multi-class attributions on ImageNet look as good as those of maximally activated neurons: attributions change significantly between classes and the highest values very visibly lie within the object of the selected class.
> >
> > While the rebuttal policy prohibits the inclusion of images, this is also hinted at by SmoothDiff's performance in *Randomization* using the *Random Logit Metric* (Sixt et al., 2020), which computes the distance between an attribution and the attribution of a randomly selected non-target class.
> > A small difference under this metric would indicate that attributions don't depend on the target class, i.e. that they are constant.
> > SmoothDiff performs well under this metric, scoring the third-highest.
> > We will include example heatmaps of multi-class attributions in the camera-ready version of the paper.

---

> > > ### Comment · Reviewer_Meht · 2025-08-06
> > > **Convinced with the Rebuttal**
> > >
> > > I would like to thank the authors for their detailed response, both the original rebuttal and the discussion phase.
> > >
> > > With the promised changes (including GridPG scores), I believe this will be an interesting paper for the community. It proposes an approach for more efficient estimation of Jacobian, and then quantitatively validates the gains. While not being the absolute SOTA attribution method, the gains of efficient estimation of Jacobian helps with raising an old method (Smooth Grad) to be on par with other baselines.
> > >
> > > I also agree with some of the points raised by other reviewers, e.g. helping to make the notation more intuitive and easier to follow. I would ask the authors to also adhere to those suggestions.
> > >
> > > I have therefore raised my score and thank the authors for their efforts!
> > >
> > > All the best!

---

> > > > ### Author Response · Authors · 2025-08-06
> > > > **Regarding the Grid Pointing Game**
> > > >
> > > > We thank the reviewer and are glad our rebuttal answered their questions.
> > > >
> > > > We have one final comment to make regarding the evaluation of the GridPG score, which is defined as the sum of positive activations within the target grid cell, normalized by the sum of all positive activations (Böhle et al., 2021, Eq. 11).
> > > > In our previous comment titled *"Regarding the localization of Integrated Gradients"*, we discussed the influence of *relevance pooling* (RP) on evaluation metrics.
> > > > Since the GridPG focuses on positive attributions, we propose to evaluate it using norm-RP on gradient-based explanations.
> > > >
> > > > For a given pixel within an input image, the gradient describes the direction in normalized RGB color space that (infinitesimally) leads to the greatest increase in the target logit.
> > > > Using a red fire truck as an example, we would expect the gradient to be positive on the red color channel (intense red positively affecting the classification) and negative on the blue and green color channels (also intensifying the red color, positively affecting the classification).
> > > > In this example, **negative values on the blue and green channels do not imply that the attribution is wrongly localized**.
> > > > Instead, gradients are closely related to difference vectors in RGB color space, motivating the use of norm-RP as a measure of sensitivity.
> > > > For the camera-ready version, we will therefore evaluate the GridPG using norm-RP for sensitivity-based methods, which is consistent with other evaluation metrics.

---

### Official Review · Reviewer_gVA9 · 2025-07-03

**Clarity:** 3
**Significance:** 3
**Originality:** 3
**Rating:** 5
**Confidence:** 4

**Summary:**

This paper proposes SmoothDiff, a method that improves upon SmoothGrad for generating gradient-based saliency maps by approximating Gaussian convolutions more efficiently. It reduces the number of required backward passes from 2n (in SmoothGrad) to just 1, while maintaining the same number of forward passes. By factorizing the Jacobian at key points (e.g., at activation functions) and using expected values of partial Jacobians, SmoothDiff achieves comparable or better attribution quality with 5–10x fewer samples than SmoothGrad. Enables substantial speed ups over standard smoothgrad.

**Questions:**

- The method approximates the expected Jacobian by ignoring cross-covariances. Can you quantify how this approximation affects attribution fidelity in practice, especially for deeper or more nonlinear architectures?

**Ethical Concerns:**

["NO or VERY MINOR ethics concerns only"]

**Final Justification:**

Overall, this paper proposes a modification to a way to compute smoothgrad by making changes to how the expected values of Jacobians is calculated. I retain my rating because the authors have adequately addressed my concerns as well as those of the other reviewers. The faithfulness metric used is not quite adequate; however, this work mostly just inherits the definition from previous work. However, this choice of metric doesn't undermine the work.

**Limitations:**

Missing Engagement with Attribution Reliability Literature
The paper does not sufficiently engage with foundational and recent research that calls into question the reliability of gradient-based attribution methods under certain conditions.

Specifically:

- Hooker et al. (2018) showed that many attribution methods, including SmoothGrad, can perform no better than random baselines unless the model satisfies specific invariance properties.

- Ilyas et al. (2019) emphasized that the problem is often not the attribution method, but the model itself, which may rely on “non-robust features” imperceptible to humans.

- Shah et al. (2021) and Srinivas et al. (2023) demonstrated that models trained for adversarial or off-manifold robustness yield much more faithful gradient-based attributions.

The authors of the present paper should acknowledge this body of work and clarify that their improvements pertain to a fixed-model regime, i.e., when the model is frozen and cannot be retrained for interpretability.

**Paper Formatting Concerns:**

None.

**Quality:**

3

**Strengths And Weaknesses:**

### Strengths
- Strong Contribution: Speed and Sample Efficiency. The paper makes a clear and compelling case that SmoothDiff improves the sample efficiency and speed of SmoothGrad by factorizing Jacobians and leveraging automatic differentiation. This is demonstrated empirically via: Faster convergence to smoothed gradients with fewer samples, High structural similarity (SSIM) with ground truth, and Performance gains in attribution evaluation tasks (e.g., pixel flipping).
- Designed to be easily integrated into existing AD frameworks (like PyTorch and JAX). Elegant use of autodiff + network structure to approximate convolution smoothing more cleverly, not just more crudely.

### Weaknesses
- However, this contribution is ultimately limited to improving the approximation of smoothgrad, and not necessarily to improving the reliability or fidelity of attributions under challenging model conditions.

---

> ### Author Rebuttal · Authors · 2025-07-31
>
> We want to thank the reviewer for their thorough feedback
> and are pleased to see the strengths of our method being recognized.
>
> Before we address the feedback, we want to highlight that due to a typo, Fig. 1b visualized SmoothGrad with the wrong sample size.
> We caught this error while working on the supplementary material and included an updated version in Appendix I.
>
> ## Weaknesses
>
> > However, this contribution is ultimately limited to improving the approximation of smoothgrad, and not necessarily to improving the reliability or fidelity of attributions under challenging model conditions.
>
> At its core, SmoothDiff efficiently approximates arbitrary integrals over gradients and Jacobians.
> Within the scope of our paper, we look at only one type of integral: the Gaussian convolution on which SmoothGrad is based.
> As we mention in our outlook, more complex integrals could be investigated, e.g. convolutions that take the geometry of the data manifold into account to improve reliability.
>
> We also want to emphasize that **both SmoothDiff and SmoothGrad are robust and reliable methods**.
> To demonstrate these properties, we have we expanded our quantitative evaluation to include five metrics proposed by Bommer et al. (2024), as well as the *Pointing Game*:
>
> * **Robustness:** Avg-Sensitivity (Yeh at el., 2019), measures the change of attributions under slight input perturbations.
> * **Faithfulness:**  Faithfulness Correlation (Bhatt et al., 2020), flips of a random subset of pixels and computes the Pearson correlation coefficient between the predicted logits and average attribution for the flipped features.
> * **Localization:**  Relevance Rank Accuracy (Arras et al., 2021), measures the ratio of high attribution pixels within a provided ground truth mask.
> * **Complexity:**  Sparseness (Chalasani et al., 2020), computes the Gini index applied to the absolute values of attributions, indicating that an attribution consists of only a few important features.
> * **Randomization:**  Random Logit Metric (Sixt et al., 2020), computes the distance between an attribution and the attribution of a randomly selected non-target class.
> * **Pointing Game:**  (Zhang et al., 2018), checks whether the point of maximal attribution lies within a provided ground truth mask.
>
> Reference implementations from Captum (Kokhlikyan et al., 2022) are used for all XAI methods except LRP, for which composites from Zennit (Anders et al., 2021) are used.
> Metrics are evaluated on a pre-trained ResNet-18 using the the Quantus toolkit (Hedström et al., 2024) with default parameters.
> For SmoothDiff, SmoothGrad, Integrated Gradients, and GradientShap, attributions are computed using 20 samples.
>
> Since this year's NeurIPS rebuttal policies prohibit the inclusion of PDFs, images and URLs, we provide our results in form of a table.
> To improve the readability of our results, we report the ranking of a method within each metric, followed by the mean score in parenthesis.
> With the exception of the *Robustness* metric, a higher score is better.
>
> **Table R1: Results on ResNet-18**
>
> |**Method**|**Robustness**|**Faithfulness**|**Localization**|**Complexity**|**Randomization**|**Pointing Game**|
> |:-|:-|:-|:-|:-|:-|:-|
> |SmoothDiff|**1 (0.45)**|3 (-0.016)|**1 (0.62)**|6 (0.430)|2 (0.51)|**1 (0.82)**|
> |SmoothGrad|2 (0.57)|2 (-0.012)|2 (0.61)|8 (0.397)|8 (0.41)|3 (0.65)|
> |Gradient|3 (0.74)|7 (-0.060)|4 (0.55)|7 (0.425)|5 (0.43)|2 (0.71)|
> |GradientShap|5 (1.19)|8 (-0.066)|7 (0.51)|5 (0.583)|6 (0.43)|3 (0.65)|
> |Integrated Gradients|4 (1.15)|6 (-0.059)|6 (0.51)|4 (0.584)|7 (0.42)|3 (0.65)|
> |Input x Gradient|6 (1.24)|5 (-0.039)|8 (0.51)|3 (0.597)|3 (0.45)|3 (0.65)|
> |LRP EpsilonPlus|7 (1.46)|4 (-0.019)|3 (0.58)|2 (0.608)|4 (0.43)|8 (0.53)|
> |LRP EpsilonAlpha2Beta1|8 (1.64)|**1 (-0.005)**|5 (0.54)|**1 (0.754)**|**1 (0.82)**|3 (0.65)|
>
> On ResNet-18, SmoothDiff has the highest *Robustness* w.r.t. input perturbations, while the high *Faithfulness* score mirrors the favorable pixel-flipping performance reported in Section 4.4.
> The *Randomization* score demonstrates that SmoothDiff attributions strongly depend on the selected output class.
> SmoothDiff performs well in the *Pointing Game* and the *Localization* metric, both of which measure the localization of attributions within ground truth masks.
> **With the exception of *Sparseness*, SmoothDiff places in the top 3 on all metrics, while being easy to implement**.
>
> ## Questions
>
> > The method approximates the expected Jacobian by ignoring cross-covariances. Can you quantify how this approximation affects attribution fidelity in practice, especially for deeper or more nonlinear architectures?
>
> We have added results on the ResNet-18 architecture in the section above, demonstrating that SmoothDiff's properties of  *Robustness*, *Faithfulness*, *Localization* and *Randomization* not only hold for sequential models (we can provide a similar results table for VGG19), but also on models with branching residual connections (see Table R1).
>
> We have also experimentally investigated ignored cross-covariances on a pre-trained VGG19 model.
> For this purpose, we compared SmoothDiff and SmoothGrad attributions (the "smoothed gradients"), by flattening them into vectors:
> *  In terms of cosine similarity, the directions of attribution vectors align well.
> * In terms of magnitude, attribution vectors differ.
> * The difference vector between attributions (which is caused by neglected cross-covariance and random sampling) therefore aligns well with the attributions
>
> During heatmapping, individual features within an attribution are normalized, such that heatmaps only depend on the direction of an attribution vector.
> Due to the directional alignment of the difference vector, we therefore observe that neglected cross-covariances don't strongly affect heatmaps, which is in line with the SSIM convergence plot in Figure 2.
>
> We will address cross-covariances in the camera-ready version by discussing these results and by adding a plot showing the convergence of attributions in terms of cosine similarity.
>
> ## Limitations
>
> > Missing Engagement with Attribution Reliability Literature The paper does not sufficiently engage with foundational and recent research that calls into question the reliability of gradient-based attribution methods under certain conditions.
>
> As mentioned above, we have added evaluation metrics according to recommendations by Bommer et al. (2024), notably
> *Avg-Sensitivity* (Yeh at el., 2019),  *Faithfulness Correlation* (Bhatt et al., 2020), *Relevance Rank Accuracy* (Arras et al., 2021), *Sparseness* (Chalasani et al., 2020), the *Random Logit Metric* (Sixt et al., 2020) and the *Pointing Game* (Zhang et al., 2018).
>
> > * Hooker et al. (2018) showed that many attribution methods, including SmoothGrad, can perform no better than random baselines unless the model satisfies specific invariance properties.
> > * Ilyas et al. (2019) emphasized that the problem is often not the attribution method, but the model itself, which may rely on “non-robust features” imperceptible to humans.
> > * Shah et al. (2021) and Srinivas et al. (2023) demonstrated that models trained for adversarial or off-manifold robustness yield much more faithful gradient-based attributions.
>
> We want to thank the reviewers for these detailed pointers.
>
> * Hooker et al. find that "SmoothGrad-Squared" (SG-SQ), a variant of SmoothGrad outperforms both SmoothGrad and other methods on their proposed ROAR benchmark. We will investigate whether an analogous "SmoothDiff-Squared" is viable for the camera-ready.
> * While we have yet to investigate SmoothDiff's behavior w.r.t. the "non-robust features" described by Ilyas et al., its performance in the **Robustness** metric of *Avg-Sensitivity* (Yeh at el., 2019) in Table R1 shows promising results.
> * We provide a comparison with $\beta$-Smoothing (Dombrowski et al., 2019) in Appendix E, which was specifically proposed as a method with off-manifold robustness. We  demonstrate that SmoothDiff outperforms $\beta$-Smoothing quantitatively in Table 1 and qualitatively in Figure 7.
>
> > [The authors should] clarify that their improvements pertain to a fixed-model regime, i.e., when the model is frozen and cannot be retrained for interpretability.
>
> We will clarify the fixed-model regime in the camera-ready version of our paper.

---

### Decision · Program_Chairs · 2025-09-17

**Decision:**

Accept (poster)

**Comment:**

The paper addresses the problem of shattered gradients in smoothgrad, a common gradient-based explanation technique, particularly for the visual domain. It provides a different approach to Gaussian kernel smoothing of smoothgrad. Particularly, it decomposes the Jacobian in a way that enables a single backward pass using automatic differentiation to obtain smoothgrad. The approach speeds up smoothgrad 5-10 times while maintaining the explanation quality.

Three knowledgeable reviewer read and evaluated the paper. All reviewers appreciated the motivation of the work, its proposed approach and the obtained empirical results evidence for the efficacy of the new method. They were also concerned about the contribution being relevant to only one one post-hoc explanation method, smoothgrad and lack of comparison with other methods.

The authors provided a thorough rebuttal which was attended by all reviewers and discussed with the authors. Particularly, they provided more comparisons with other methods as requested by the reviewers. All reviewers, eventually, suggest acceptance.

The AC believes smoothgrad is a relatively common posthoc explanation method with use cases in application fields. The paper makes a clear and effective contribution to that which makes it comparable or better to other existing posthoc methods. This is enough contribution and therefore the AC suggests acceptance.